# Exploring the Limits of Differentially Private Deep Learning with Group-wise Clipping

**Jiyan He**[1†]**, Xuechen Li**[2†]**, Da Yu**[3†]**, Huishuai Zhang**[4]**, Janardhan Kulkarni**[4]**,**
**Yin Tat Lee**[4]**, Arturs Backurs**[4]**, Nenghai Yu**[1]**, Jiang Bian**[4*]

[1]University of Science and Technology of China, [2]Stanford University, [3]Sun Yat-sen University
[4]Microsoft Research,
{hejiyan@mail,ynh@}.ustc.edu.cn, lxuechen@cs.stanford.edu,
yuda3@mail2.sysu.edu.cn, {huzhang,jiang.bian}@microsoft.com,
{jakul,yintatlee,arturs.backurs}@microsoft.com.

## Abstract

Differentially private deep learning has recently witnessed advances in computational efficiency and privacy-utility trade-off. We explore whether further improvements along the two axes are possible and provide affirmative answers leveraging two instantiations of *group-wise clipping*. To reduce the compute time overhead of private learning, we show that *per-layer clipping*, where the gradient of each neural network layer is clipped separately, allows clipping to be performed in conjunction with backpropagation in differentially private optimization. This results in private learning that is as memory-efficient and almost as fast per training update as non-private learning for many workflows of interest. While per-layer clipping with constant thresholds tends to underperform standard flat clipping, per-layer clipping with adaptive thresholds matches or outperforms flat clipping under given training epoch constraints, hence attaining similar or better task performance within less wall time. To explore the limits of scaling (pretrained) models in differentially private deep learning, we privately fine-tune the 175 billion-parameter GPT-3. We bypass scaling challenges associated with clipping gradients that are distributed across multiple devices with *per-device clipping* that clips the gradient of each model piece separately on its host device. Privately fine-tuning GPT-3 with per-device clipping achieves a task performance at $\epsilon = 1$ better than what is attainable by non-privately fine-tuning the largest GPT-2 on a summarization task.

## 1 Introduction

Recent works on deep learning with differential privacy (DP) have substantially improved the computational efficiency (Subramani et al., 2021; Anil et al., 2021) and privacy-utility trade-off (Li et al., 2022a; Yu et al., 2022; De et al., 2022; Mehta et al., 2022), resulting in cost-effective private learning workflows with favourable utility under common levels of privacy guarantee. Common to most of these works is the use of differentially private stochastic gradient descent (DP-SGD) which clips per-example gradients (herein referred to as *flat clipping*) and noises their average before performing the parameter update based on a minibatch (Song et al., 2013; Bassily et al., 2014; Abadi et al., 2016). We explore whether further improvements in computational efficiency and privacy-utility trade-off are possible and provide affirmative answers for both directions, leveraging two instantiations of *group-wise clipping* for DP-SGD.

DP-SGD is known to be computationally costly due to clipping per-example gradients. Instantiating per-example gradients and (potentially) normalizing them can incur both high memory and time costs in standard machine learning frameworks (Paszke et al., 2019; Frostig et al., 2018), and thus private machine learning with DP-SGD is reportedly much more memory demanding and/or slower than its non-private counterpart (Carlini et al., 2019; Hoory et al., 2021). Recent works have considerably improved the memory and time efficiency of DP-SGD with better software primitives (Subramani et al., 2021) and algorithms (Yousefpour et al., 2021; Lee & Kifer, 2021; Li et al., 2022b; Bu et al.,

---

[*]The work of Jiyan He, Xuechen Li and Da Yu was done while they were interns at Microsoft Research. †
Equal contributions. Correspondence to Jiyan He, Xuechen Li, Huishuai Zhang and Janardhan Kulkarni.

2022). Nevertheless, private learning still shows non-trivial increases in either memory usage or compute time when compared to non-private learning head-to-head. For instance, better software primitives do not eliminate the inherent increase in memory spending (Subramani et al., 2021), and improved algorithms only remove this memory overhead at the cost of extra runtime (Li et al., 2022b). The first research question we study is therefore

*Can private learning be as memory and time efficient (per epoch) as non-private learning?*

We answer the above question affirmatively by giving an efficient implementation of *per-layer clipping* which had been studied in past works but not from a computational efficiency perspective (McMahan et al., 2018b; Dupuy et al., 2022). Clipping per-example gradients of separate neural networks layers (e.g., linear, convolution) separately allows clipping to be performed in conjunction with backpropagation. This results in private learning that is as memory-efficient and almost as time-efficient per training update as non-private learning for many small to moderate scale workflows of practical interest. While per-layer clipping with static clipping thresholds chosen by hand tends to underperform flat clipping, we show that per-layer clipping with adaptively estimated thresholds matches or outperforms flat clipping under given training epoch constraints, hence attaining similar or better task performances with less wall time.

DP-SGD is known to (possibly) incur substantial performance losses compared to non-private learning. To improve the privacy-utility trade-off, several past works have leveraged large-scale publicly pretrained models (Yu et al., 2022; Li et al., 2022b; De et al., 2022; Mehta et al., 2022). These works observe that the privacy-utility trade-off improves with the use of larger (and thus better) pretrained models.[1] We extend this research and study a second research question

*Can the privacy-utility trade-off be further improved with even better / larger pretrained models?*

To study this, we scale DP fine-tuning to work with one of the largest and most performant pretrained language models to date—the original 175 billion-parameter GPT-3. Weights of this model cannot be hosted in the memory of a single GPU and must be distributed across multiple devices. This presents challenges for flat clipping which calls for communicating per-example gradient norms across devices. To bypass these challenges, we turn to *per-device clipping*, where each device is prescribed a clipping threshold for clipping per-example gradients of the hosted model piece. Per-device clipping incurs no additional communication cost and allowed us to obtain with GPT-3 a private fine-tuning performance at $\epsilon = 1$ that is better than what is attainable by non-privately fine-tuning the largest GPT-2 on a challenging summarization task. Our contributions are summarized below.

(1) We show per-layer clipping enables clipping to be done in conjunction with backpropagation in DP optimization and results in private learning that is as memory-efficient and almost as fast per training update as non-private learning for many small to moderate scale workflows of interest.

(2) We show adaptive per-layer clipping matches or outperforms flat clipping under fixed training epoch constraints, and thus attains similar or better task performances with less wall time.

(3) We bypass scaling challenges associated with communicating per-example gradient norms with per-device clipping, with which we scale DP fine-tuning to work with the 175 billion-parameter GPT-3 and obtain improved task performance for a challenging summarization task at $\epsilon = 1$.

## 2 PRELIMINARIES

This section aims to cover background on gradient clipping in DP optimization and explain why alternate group-wise clipping strategies can be attractive from a computational efficiency standpoint.

Our work trains machine learning models (more precisely deep neural networks) with optimizers that guarantee DP. For completeness, we recap the definition of approximate DP/$(\epsilon, \delta)$-DP below.

**Definition 2.1** (($\epsilon, \delta$)-DP)**.** A randomized algorithm $\mathcal{M} : \mathcal{X} \to \mathcal{Y}$ is $(\epsilon, \delta)$-DP if for all neighboring datasets $\mathbb{D}, \mathbb{D}' \in \mathcal{X}$ and all $Y \subset \mathcal{Y}$, it holds that $\mathbb{P}\left(\mathcal{M}(\mathbb{D}) \in Y\right) \leq \exp(\epsilon) \cdot \mathbb{P}\left(\mathcal{M}(\mathbb{D}') \in Y\right) + \delta$.

---

[1]While size does not equate quality, there is strong correlation between the two under currently popular pretraining techniques (Liu et al., 2019; Brown et al., 2020). We're optimistic that future smaller models pretrained with improved techniques can be as performant as current large models (Hoffmann et al., 2022).

In our case, $\mathcal{M}$ is an optimization algorithm which outputs the learned model parameters $\boldsymbol{\theta}$. Widely used DP optimizers (e.g., DP-SGD) usually introduce two additional steps before each parameter update to privatize gradients: (1) clip per-example gradients of a minibatch by their Euclidean norms according to some threshold $C$; (2) add Gaussian noise to the sum of clipped gradients. In practice, the clipping threshold $C$ can be either a tunable hyperparameter or set to some (privatized) statistic estimated from data. The standard deviation of the Gaussian noise is determined by the clipping threshold $C$ and the noise multiplier $\sigma$, the latter of which is set by a privacy accounting procedure given target privacy parameters $(\epsilon, \delta)$, the number of iterations $T$, and the subsampling rate $\rho$ (Abadi et al., 2016; Mironov, 2017; Dong et al., 2021; Gopi et al., 2021). We now present background on different per-example gradient clipping strategies in DP optimization.

**Flat Clipping.** This is the clipping scheme used in the original DP-SGD algorithm (Abadi et al., 2016). Here, the gradient of example $s_i$'s loss $\ell(\boldsymbol{\theta}, s_i)$ with respect to model parameters $\boldsymbol{g}^{(i)} := \partial\ell(\boldsymbol{\theta}, s_i)/\partial\boldsymbol{\theta}$ is normalized if its magnitude exceeds the threshold $C$. Thus, the actual contribution (up to scaling) of the $i$th instance to the noisy gradient is $\widetilde{\boldsymbol{g}}^{(i)} := \boldsymbol{g}^{(i)} \cdot \min\{1, C/\|\boldsymbol{g}^{(i)}\|\}$. Flat clipping cannot be performed until the gradient norms $\{\|\boldsymbol{g}^{(i)}\|\}_i$ are computed. Since the latter quantities are only known after backpropagation completes, flat clipping necessitates a second-round of computation after backpropagation to conditionally rescale the gradients. This is a source of overhead and presents complications when model weights don't fit on a single device.

**Group-Wise Clipping.** This scheme partitions the set of parameters $\boldsymbol{\theta} \in \mathbb{R}^d$ into $K$ disjoint groups $\{\boldsymbol{\theta}_k\}_{k=1}^K$ with $\boldsymbol{\theta}_k \in \mathbb{R}^{d_k}$ for $k \in [K]$. For each group $k$, the scheme prescribes a clipping threshold $C_k$. Denote example $s_i$'s gradient for the $k$th group by $\boldsymbol{g}_k^{(i)} := \partial\ell(\boldsymbol{\theta}, s_i)/\partial\boldsymbol{\theta}_k$. Under group-wise clipping, the $k$th clipped gradient for $s_i$ is $\tilde{\boldsymbol{g}}_k^{(i)} := \boldsymbol{g}_k^{(i)} \cdot \min\{1, C_k/\|\boldsymbol{g}_k^{(i)}\|\}$. Next, we present two instantiations of group-wise clipping that are computationally advantageous in different settings.

## 3 EFFICIENT PRIVATE LEARNING WITH ADAPTIVE PER-LAYER CLIPPING

The first instantiation of group-wise clipping we study is per-layer clipping which clips gradients of separate neural network layers separately. This scheme had been presented in past works (McMahan et al., 2018a;b; Dupuy et al., 2022), but neither its computational properties nor its performance implications had been carefully studied. We show that with proper implementation, per-layer clipping can be as memory-efficient and almost as time-efficient as non-private training for small- to moderate-scale workflows that run on single accelerators. Additionally, we confirm that per-layer clipping with hand-set thresholds underperforms flat clipping and demonstrate that adaptively setting these thresholds eliminates potential performance losses.

### 3.1 PER-LAYER CLIPPING DP-SGD CAN BE ALMOST AS EFFICIENT AS NON-PRIVATE SGD

Per-layer clipping groups together parameters of a neural network layer (e.g., linear, convolution) and prescribes each of the $K$ layers of the network a clipping threshold $C_k$ to clip the gradient of that layer. This directly implies that gradient clipping for any layer can be performed as soon as backpropagation reaches that layer (to construct per-example gradients or norms) when parameter sharing is absent, and is unlike flat clipping which cannot be performed until backpropagation completes entirely.[2]

Our efficient implementation of per-layer clipping clips layer-wise gradients as soon as the gradient with respect to outputs of that layer are returned from backpropagation. The operations of clipping and summing per-example gradients can be fused once input activations, output gradients, and per-example gradient norms are known. In addition, per-example gradient norms can be cheaply computed without materializing actual per-example gradients in memory (Li et al., 2022b, Section 4). This implementation results in private training that is as memory-efficient as non-private training since per-example gradients are not instantiated, and almost as time-efficient per update since the extra computation involving gradient norm and gradient scaling are typically cheap. Figure 1 shows that carefully implemented per-layer clipping matches the memory profile and almost matches the training throughput of non-private learning for an autoregressive fine-tuning task with GPT-2 on a single GPU (we followed the same experimental protocol as that of Section 4 in (Li et al., 2022b) for a fair comparison). See Appendix G for experiments with head-to-head wall time comparisons.

---

[2]Note the DP learning library Opacus (Yousefpour et al., 2021) has a per-layer clipping optimizer which supports clipping each layer with a separate threshold. But this implementation conducts per-layer clipping after backpropagation completes and instantiates all per-example gradients before clipping, which is inefficient.

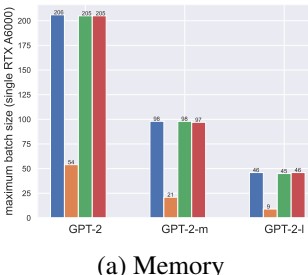

(a) Memory

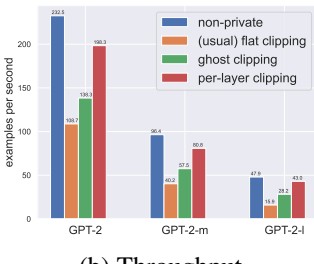

(b) Throughput

Figure 1: Private learning with (adaptive) per-layer clipping can be almost as efficient as non-private learning (the throughput gap is less than 15% in this case). Here, "(usual) flat clipping" refers to the implementation which first creates and stores in memory all per-example gradients (e.g., adopted in Opacus (Yousefpour et al., 2021)). Ghost clipping (Li et al., 2022b) is based on flat clipping but avoids materializing per-example gradients by performing an additional backward pass each time.

## 3.2 FIXED PER-LAYER CLIPPING MAY HURT THE UTILITY

Despite the computational advantages, per-layer clipping with fixed clipping thresholds set by hand (which we refer to as *fixed per-layer clipping*) reportedly underperforms flat clipping (McMahan et al., 2018b). To further verify this and remove confounding effects of potentially suboptimal hyperparameters, we compare fixed per-layer clipping against (fixed) flat clipping on two tasks: 1) training wide ResNet (WRN16-4) (Zagoruyko & Komodakis, 2016) from scratch to classify CIFAR-10 images, and 2) fine-tuning the pretrained RoBERTa-base for classifying sentiment on SST-2. We carefully tuned the clipping thresholds and learning rate for both clipping methods; see Appendix A for details. Tables 1a and 1b confirm that per-layer clipping with hand-set fixed thresholds underperforms flat clipping.

Table 1: Fixed per-layer clipping underperforms (fixed) flat clipping.

(a) CIFAR-10 validation accuracy over 3 seeds.

| Model | Method | $\epsilon = 3$ | $\epsilon = 8$ |
|---|---|---|---|
| WRN16-4 | Fixed per-layer | 60.6 | 67.8 |
| | Fixed flat | 63.1 | 73.9 |

(b) SST-2 validation accuracy over 3 seeds.

| Model | Method | $\epsilon = 3$ | $\epsilon = 8$ |
|---|---|---|---|
| RoBERTa-base | Fixed per-layer | 89.4 | 89.7 |
| | Fixed flat | 91.0 | 91.7 |

To understand why fixed per-layer clipping gives worse performance, we plot the per-layer gradient norms of randomly sampled CIFAR-10 examples for privately training WRN16-4 in Figure 2 (see Appendix B for the setup). We observe that the general magnitudes of per-layer gradient norms change dramatically across training. Early on, gradient norms are generally uniformly low across all layers. As training proceeds, gradient norms for layers close to the input gradually become high. We present additional evidence for this phenomenon with language model fine-tuning in Appendix B.

These observations suggest that clipping with fixed layer-wise thresholds likely removes the structural relation between gradients of different layers. This incurs an extra source of bias in addition to the usual bias of flat clipping that alters the relation of gradients across samples, and makes balancing clipping bias and privacy noise throughout training more challenging. These observations motivate us to set the thresholds based on some adaptively estimated statistic of layer-wise gradients.

## 3.3 ADAPTIVE PER-LAYER CLIPPING CAN BE AS EFFECTIVE AS FLAT CLIPPING

To overcome the performance issues of fixed per-layer clipping, we consider per-layer clipping with adaptive clipping thresholds that we herein refer to as *adaptive per-layer clipping*. Our hope is that the adaptive thresholds can track gradient norm shift, capture gradient structure, and consequently mitigate the structural bias caused by clipping gradients of separate layers separately. One candidate statistic for setting the adaptive threshold is some quantile of gradient norms. Notably, Andrew et al. (2019) provided an effective way of estimating quantiles privately for flat clipping via online convex optimization. We adapt their algorithm to the per-layer setup, and let each layer maintain an online estimate of a target gradient norm quantile.

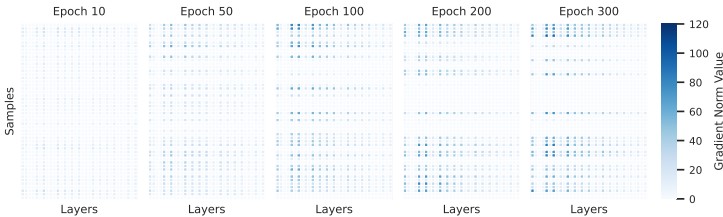

Figure 2: The distribution of per-layer gradient norms shifts substantially across training. Each column represents one layer, and each row represents one example. Layers of the neural network are placed from input (left) to output (right). Darker colors indicate higher values of gradient norms.

Two questions arise with this formulation: 1) How should the per-layer gradient norm quantiles be privately estimated, and 2) how should the noise levels for different layers be decided? We address the two questions below. Algorithm 1 delineates the overall procedure, where per-layer gradient clipping in conjunction with backpropagation occurs on lines 7-12, adaptive private quantile estimation occurs on lines 15-18, and noise allocation occurs on line 13. The pseudocode is based on DP-SGD, but its core ideas naturally apply to private versions of other first-order optimizers (e.g., DP-Adam).

**Estimating Quantiles Privately.** We allocate some privacy budget (in practice $r = 1\%$ to $10\%$ of total budget) to estimate a target quantile of each layer's gradient norms. Clipping thresholds $C_1, ..., C_K$ are then set to these estimated quantiles. We record the number of gradients clipped before each parameter update and adjust the clipping threshold based on whether too many or too few are clipped. The central quantity which needs to be privatized is then the fraction of clipped gradients. We introduce the additional noise multiplier $\sigma_b$ to privatize this fraction statistic (used in Gaussian mechanism). The new noise multiplier $\sigma_{\text{new}}$ (based on $1 - r$ fraction of total budget) for noising parameter updates is computed with the following proposition, whose proof we defer to Appendix D.

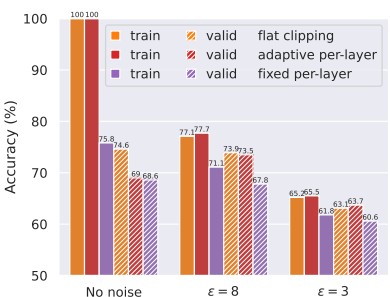

Figure 3: Adaptive per-layer clipping eliminates performance losses experienced by fixed per-layer clipping.

**Proposition 3.1.** *Let $\sigma$ be the original noise multiplier for noising parameter updates to achieve a certain level of differential privacy (without private quantile estimation) and $\sigma_b$ be chosen for noising quantile estimates (release of the latter consumes $r$ fraction of the privacy budget). Then the new noise multiplier $\sigma_{new}$ for noising parameter updates (consuming $1 - r$ fraction of the budget) is*

$$\sigma_{new} = (\sigma^{-2} - K/(2\sigma_b)^2)^{-1/2}. \tag{3.1}$$

*Remark* 3.1. The private quantile estimation for $K$ groups costs a fraction $r = K\sigma^2/(4\sigma_b^2)$ of privacy budget (in terms of Rényi differential privacy (Mironov, 2017)).

**Allocating Noise.** The original Gaussian mechanism adds isotropic noise to statistics before their release (Dwork et al., 2014) which results in different coordinates experiencing the same amount of noise. Yet, simply scaling different components with public quantities (before adding noise) allows different components to experience different levels of noise. As an example, let $\gamma_1, \cdots, \gamma_K$ be coefficients for scaling, and recall that $\tilde{g}_k$ is the sum of clipped gradients for layer / group $k$. Then, applying the Gaussian mechanism to the scaled $\hat{g} := (\hat{g}_1, ..., \hat{g}_K)$, where $\hat{g}_k := \tilde{g}_k/\gamma_k$, and rescaling back the privatized quantities afterwards ends up adding noise to $\tilde{g}_k$ that has standard deviation proportional to $\gamma_k$.

Among the possible ways of choosing $\{\gamma_1, ..., \gamma_K\}$, we outline two simple approaches that we found to be effective for different reasons in our empirical studies. We use the global strategy in all but experiments with GPT-3. Appendix E includes empirical studies of alternate strategies.

- *Global strategy*: $\gamma_k = 1$ for $k \in [K]$. This strategy adds the same amount of noise to every component. The total noise has squared $\ell_2$ norm $V_G \propto (\sum_k C_k^2) \cdot (\sum_k d_k)$.
- *Equal budget strategy*: $\gamma_k = C_k$ for $k \in [K]$. Each group has the same amount of privacy budget. The total noise has squared $\ell_2$ norm $V_E \propto K \sum_{k=1}^{K} d_k C_k^2$.

---

**Algorithm 1** DP-SGD with adaptive per-layer clipping

---

1: **INPUT**: Private dataset $\mathbb{D} = \{s_i\}_{i=1}^N$; initial iterate $\boldsymbol{\theta}_0$; number of iterations $T$; learning rate $\eta_t$; learning rate for quantile estimation $\eta$; privacy parameters $\epsilon, \delta$; per-layer parameters $\{\boldsymbol{\theta}_1, \ldots, \boldsymbol{\theta}_K\}$; initial clipping thresholds $\{C_1, \ldots, C_K\}$; weighting factors $\{\gamma_1, \ldots, \gamma_K\}$; target quantile $q$; sampling rate $\rho = B/N$.
2: $\sigma \leftarrow \text{PrivacyAccountant}(\epsilon, \delta, \rho, T)$
3: Choose $\sigma_b$ as the noise multiplier for private quantile estimation
4: Compute the new noise multiplier $\sigma_{\text{new}}$ for gradient privatization with equation 3.1
5: **for** $t = 0$ to $T - 1$ **do**
6:     Sample a minibatch $\mathcal{S}_t$ with sampling rate $\rho$ and perform the forward pass
7:     **for** $k = K$ to $1$ **do**
8:         Compute per-sample gradient norms $\{\|\boldsymbol{g}_k^{(i)}\|\}_{i \in \mathcal{S}_t}$ given activations and output gradients
9:         Compute $\tilde{\boldsymbol{g}}_k \leftarrow \sum_{i \in \mathcal{S}_t} \tilde{\boldsymbol{g}}_k^{(i)} = \sum_{i \in \mathcal{S}_t} \boldsymbol{g}_k^{(i)} \cdot \min\{1, C_k/\|\boldsymbol{g}_k^{(i)}\|\}$ with fused operation
10:         Record $\bar{b}_k \leftarrow \sum_{i \in \mathcal{S}_t} \mathbb{1}[\|\boldsymbol{g}_k^{(i)}\| \leq C_k]$ for quantile estimation
11:         Perform usual backpropagation to obtain input gradients if $k > 1$
12:     **end for**
13:     Draw $\boldsymbol{z} \leftarrow (\boldsymbol{z}_1, \ldots, \boldsymbol{z}_K)$, where $\boldsymbol{z}_k \sim \mathcal{N}\left(0, \sigma_{\text{new}}^2 S^2 \gamma_k^2 \boldsymbol{I}_{d_k}\right)$ and $S = (\sum_{k=1}^K C_k^2/\gamma_k^2)^{1/2}$
14:     $\boldsymbol{\theta}_{t+1} \leftarrow \boldsymbol{\theta}_t - \eta_t (\tilde{\boldsymbol{g}} + \boldsymbol{z})/B$.
15:     **for** $k = 1$ to $K$ **do**
16:         Draw $z_k \sim \mathcal{N}(0, \sigma_b^2)$, set $\tilde{b}_k \leftarrow (\bar{b}_k + z_k)/B$, and $C_k \leftarrow C_k \cdot \exp(-\eta(\tilde{b}_k - q))$.
17:     **end for**
18: **end for**
19: **return** $\boldsymbol{\theta}_T$ (or some average of all iterates)

---

With the tools of quantile estimation and noise allocation, we show that adaptive per-layer clipping matches the performance of flat clipping. . Figure 3 compares adaptive per-layer clipping against fixed per-layer clipping and flat clipping with and without noise for training WRN16-4 on CIFAR-10 (details in Appendix A), and shows that the performance of adaptive per-layer clipping matches that of flat clipping, while fixed adaptive clipping suffers large performance drops. Section 5 includes additional results to validate this point.

## 4 EFFICIENT PRIVATE PIPELINE PARALLELISM WITH PER-DEVICE CLIPPING

Past works have shown that DP fine-tuning yields improved privacy-utility trade-offs with the use of larger / better pretrained models. We study whether this trend continues to hold as one leverages larger pretrained models by scaling DP training to work with one of the largest pretrained language models to date—the 175 billion-parameter GPT-3. The sheer size of this model presents challenges in computational efficiency, since model weights cannot be fit on a single device (e.g., GPU) and existing approaches for distributing computation don't tend to play well with flat clipping.

We base our distributed DP training strategy off the popular *pipeline parallelism* used in non-private training (Huang et al., 2019; Rasley et al., 2020).[3] We summarize the idea of pipeline parallelism here and defer to the cited works for the specifics. Pipeline parallelism first partitions the model into chunks of consecutive layers / blocks and distributes each onto a single accelerator. Forward computation with a microbatch (created through splitting a minibatch) then chains together local computations with each model piece (hosted on each accelerator) by communicating activations across accelerators. Backward computation (backpropagation) roughly reverses the above process, but on each accelerator, intermediate forward activations of the model piece are recomputed to reduce peak memory (Huang et al., 2019, Section 2.3). Most importantly, pipeline parallelism simultaneously performs computation with different microbatches on different accelerators to reduce the overall idle time. Devices synchronize after all microbatches finish their forward and backward computation and before the optimizer invokes the parameter update.

Flat clipping necessitates computing per-example gradient norms to correctly rescale gradients. This calls for the communication of per-example norms of local gradients on each device and leads to an

---

[3]Alternate parallelization schemes can be more flat clipping friendly (e.g., FSDP (Zhao et al., 2022)), but current open source implementations of these schemes are generally not light-weight fine-tuning friendly.

inherent overhead in pipeline parallelism. We outline two potential approaches for accomplishing communication, both of which unfortunately lead to non-trivial slowdowns as well as complications in implementation. The first approach synchronizes all devices after the full backward pass finishes for each microbatch (within a minibatch) so that each device will retain the same gradient norms for computing the scaling factor in clipping. This approach incurs as many extra synchronization steps as the number of microbatches per minibatch and reduces training efficiency when the number of microbatches is large. While executing primitives like all-gather with local gradient norms is not costly per se, the disruption these calls bring to the pipeline schedule is. Concretely, devices need to perform one of the following: (i) retain the unclipped local per-example gradients for a microbatch—and become idle due to pausing its processing of subsequent microbatches to avoid memory errors—until synchronization for the microbatch is called; (ii) offload the unclipped local per-example gradients to CPU only to transport them back on synchronization; (iii) rematerialize the microbatch's local gradient on synchronization. (i) is costly since it forces devices to be idle, (ii) is costly due to slow CPU-GPU data transfer, and (iii) is costly due to performing the extra round of backpropagation. To reduce the frequency of synchronization, a second approach may instead ask devices to only synchronize after the last microbatch has been processed. This approach, however, does not bypass the complications in the subsequent gradient rescaling step which requires local per-example gradients either be offloaded to CPU and moved back later or rematerialized on synchronization.

As the first attempt at experimenting with DP fine-tuning on huge models, we instead turn to an alternative *per-device clipping* scheme, where each device is prescribed a clipping threshold for clipping per-example gradients of the hosted model piece. Leveraging the equal budget strategy, the noise level added to gradients on each device is agnostic of the clipping thresholds of other devices (thus, no extra communication incurred). We present the full pseudocode of the algorithm in Appendix C. Notably, per-device clipping with DP LoRA fine-tuning allowed us to obtain improved results for a challenging summarization task (see Section 5.3).

## 5 EXPERIMENTS

Previous sections verified that per-layer clipping has an efficiency advantage over flat clipping. We now show that adaptive per-layer clipping is competitive in terms of privacy vs utility. Our experiments cover training wide ResNets from scratch, fine-tuning RoBERTa on GLUE tasks, and fine-tuning GPT-2 and GPT-3 for table-to-text generation and summarization tasks. For private quantile estimation, we use the geometric update rule by Andrew et al. (2019) and set $\eta = 0.3$ for all experiments. Reported numbers are averaged over three seeds unless otherwise stated. Code to reproduce some of our experiments can be found at `https://github.com/lxuechen/perlayer-public`.

### 5.1 PRIVATELY LEARN WIDERESNETS FOR CIFAR-10 CLASSIFICATION

We train a wide ResNet (WRN16-4, 2.8M trainable parameters) (Zagoruyko & Komodakis, 2016) from scratch for CIFAR-10 classification with differential privacy. We follow the implementation by De et al. (2022), e.g., batch normalization are replaced with group normalization and weight standardization is applied for convolutional layers, except that we do not use augmentation multiplicity for simplicity. We set privacy parameter $\delta = 10^{-5}$ and choose $\epsilon$ from $\{1, 3, 5, 8\}$, which are typical privacy parameters used in previous works.

We compare the performance of adaptive per-layer clipping with that of flat clipping. For both algorithms, we use hyperparameters suggested by De et al. (2022) and tune learning rates. We use a fraction $r = 0.01$ of privacy budget for quantile estimation and choose the target quantile $q$ from $\{0.5, 0.6, 0.7\}$. For both algorithms we train for 300 epochs. We summarize the details in Appendix A.1. Table 2 shows that adaptive per-layer clipping achieves training and validation accuracies on par with flat clipping for multiple choices of $\epsilon$.

Table 2: Adaptive per-layer clipping achieves accuracy (in %) on par with flat clipping for CIFAR-10.

| Method | $\epsilon = 1$ | | $\epsilon = 3$ | | $\epsilon = 5$ | | $\epsilon = 8$ | |
|---|---|---|---|---|---|---|---|---|
| | Train | Valid | Train | Valid | Train | Valid | Train | Valid |
| Flat clipping (De et al., 2022) | 44.9 | 44.7 | 65.2 | 63.1 | 72.3 | 70.1 | 77.1 | 73.9 |
| Adaptive per-layer | 49.1 | 48.6 | 65.5 | 63.7 | 72.3 | 69.5 | 77.7 | 73.5 |

## 5.2 PRIVATELY FINE-TUNE ROBERTA ON GLUE TASKS

Our first experiment aims to show that adaptive per-layer clipping is broadly competitive with existing approaches in the literature in terms of the privacy-utility tradeoff. In this experiment, we fine-tune RoBERTa-base (125M) and RoBERTa-large (355M) (Liu et al., 2019) on SST-2, QNLI, QQP, and MNLI from the GLUE benchmark (Wang et al., 2018) with differential privacy. We set $\epsilon \in \{3, 8\}$ and $\delta = 1/n^{1.1}$, where $n$ is the size of training set. We tune the learning rate, batch size, and target quantile on SST-2's training data and transfer the best hyperparameters to other tasks. We use $r = 0.1$ of the privacy budget for quantile estimation, choose the target quantile $q$ from $\{0.5, 0.75, 0.85\}$, and set the number of training epochs $E = 20$. Table 3 shows that adaptive per-layer clipping obtains accuracies competitive with established approaches in the literature under fixed privacy constraints.

Our second controlled experiment shows that adaptive per-layer clipping gives utility that is competitive with flat clipping under fixed training epochs (when both approaches fine-tune the same set of parameters), effectively verifying its wall time advantage. In this experiment, we constrain the number of training epochs $E$ to be one of $\{3, 10, 20, 30\}$ and fine-tune RoBERTa models on SST-2 with the two clipping methods. Table 4 shows that adaptive per-layer clipping is on par with flat clipping in accuracy for all setups and justifies the wall time advantage claim given that adaptive per-layer clipping is faster per epoch.

Li et al. (2022b) demonstrated that the performance of private fine-tuning for text classification with various algorithms improves with a text infilling formulation. The infilling technique reformulates the optimization problem and is orthogonal to the algorithmic aspects under study. To ensure our comparisons are fair, all experiments in this section follow the usual BERT fine-tuning setup without text infilling. Additional details of the two experiments can be found in Appendix A.2.

Table 3: Adaptive per-layer clipping matches accuracy (in %) results in the literature on GLUE tasks.[4]

| Model | Method | $\epsilon = 3$ | | | | $\epsilon = 8$ | | | |
|---|---|---|---|---|---|---|---|---|---|
| | | MNLI-(m/mm) | QQP | QNLI | SST-2 | MNLI-(m/mm) | QQP | QNLI | SST-2 |
| RoBERTa-base | Yu et al. (2021b) | - | - | - | - | 80.1 | 85.5 | 87.2 | 91.6 |
| | Li et al. (2022b) | 82.47/82.10 | 85.41 | 84.62 | 86.12 | 83.30/83.13 | 86.15 | 84.81 | 85.89 |
| | Yu et al. (2022) | - | - | - | - | 83.5 | 85.7 | 87.3 | 92.2 |
| | Adaptive per-layer | 82.83/83.27 | 85.67 | 86.13 | 92.03 | 83.70/83.97 | 86.23 | 87.13 | 92.40 |
| RoBERTa-large | Yu et al. (2021b) | - | - | - | - | 86.1 | 86.7 | 90.0 | 93.0 |
| | Li et al. (2022b) | 85.53/85.81 | 86.65 | 88.94 | 90.71 | 86.28/86.54 | 87.49 | 89.42 | 90.94 |
| | Yu et al. (2022) | - | - | - | - | 87.8 | 87.4 | 90.8 | 95.3 |
| | Adaptive per-layer | 87.10/87.20 | 86.80 | 89.80 | 93.87 | 87.67/87.57 | 87.20 | 90.77 | 94.03 |

Table 4: Adaptive per-layer clipping is competitive with flat clipping on SST-2 in accuracy (%) under fixed fine-tuning epochs ($E$). Models are full fine-tuned. Numbers in parentheses are standard deviations from three independent runs. See results for $\epsilon = 8$ in Appendix H.

| Model | Method | $\epsilon = 3$ | | | |
|---|---|---|---|---|---|
| | | $E = 3$ | $E = 10$ | $E = 20$ | $E = 30$ |
| RoBERTa-base | Flat clipping (tuned) | 88.70(0.52) | 90.17(1.10) | 91.47(1.07) | 91.60(0.95) |
| | Adaptive per-layer | 90.50(1.21) | 91.90(0.72) | 92.33(0.42) | 92.23(0.06) |
| RoBERTa-large | Flat clipping (tuned) | 92.20(0.26) | 93.07(0.75) | 93.67(0.40) | 94.23(0.67) |
| | Adaptive per-layer | 91.73(0.59) | 93.27(0.45) | 93.90(0.26) | 94.13(0.38) |

## 5.3 PRIVATELY FINE-TUNE ON LANGAUGE GENERATION TASKS

**Table-To-Text Generation.** We compare adaptive per-layer clipping against flat clipping for full fine-tuning with GPT-2 on the E2E (Novikova et al., 2017) and DART (Nan et al., 2020) table-to-text generation tasks. Since Li et al. (2022b) performed extensive tuning on these tasks for flat clipping, we recall their results here. For runs with adaptive per-layer clipping, we reused hyperparameter values tuned for SST-2, but re-tuned the target quantile with the E2E

---

[4]Results in Yu et al. (2021b; 2022) on MNLI are the average of the matched and mismatched accuracy.

Table 5: Adaptive per-layer clipping matches the performance of flat clipping for full fine-tuning under the same number of training epochs for common privacy levels. Results based on fine-tuning GPT-2 on E2E and DART. Numbers in the column "flat" are reported by Li et al. (2022b).

| Metric | DP Guarantee | E2E | | DART | |
|---|---|---|---|---|---|
| | | Adaptive per-layer | Flat | Adaptive per-layer | Flat |
| BLEU | $\epsilon = 3$ | 61.101 | 61.519 | 31.851 | 31.025 |
| | $\epsilon = 8$ | 63.416 | 63.189 | 34.166 | 35.057 |
| | non-private | - | 69.463 | - | 42.783 |
| ROUGE-L | $\epsilon = 3$ | 65.120 | 65.670 | 51.519 | 52.063 |
| | $\epsilon = 8$ | 66.689 | 66.429 | 52.877 | 54.576 |
| | non-private | - | 71.359 | - | 56.717 |

validation set and constrained the batch size and number of training epochs to be the same as that for flat clipping; see Appendix A.3 for details. Table 5 confirms that DP learning with adaptive per-layer clipping performs comparably to flat clipping under a given epoch constraint.

**Dialog Summarization.** We use the SAMSum dialog summarization task as a testbed for studying model scaling (Gliwa et al., 2019).[5] This task is more challenging than previously tested ones since its training set is small (less than 15k examples) and inputs are long. We fine-tune both GPT-2-xl and the (original) 175 billion-parameter GPT-3 with LoRA (Hu et al., 2021) with and without DP, and compare them against in-context learning with GPT-3 (Brown et al., 2020). Table 6 shows that GPT-3 fine-tuned at $\epsilon = 1$ outperforms non-privately fine-tuned GPT-2-xl and in-context learning with 4 demonstrations (the maximum that can be fitted within the context window of 2048 tokens). See Appendix C for more details.

Table 6: Fine-tuning GPT-3 with DP LoRA achieves improved privacy-utility trade-off. Metrics are ROUGE scores.

| Model+Method+DP guarantee | R-1 | R-2 | R-L |
|---|---|---|---|
| **Flat clipping** | | | |
| GPT-2-xl LoRA ($\epsilon = 0.25$) | 8.0 | 2.7 | 6.8 |
| GPT-2-xl LoRA ($\epsilon = 1$) | 30.0 | 11.0 | 25.3 |
| GPT-2-xl LoRA ($\epsilon = 4$) | 35.4 | 14.3 | 29.9 |
| GPT-2-xl LoRA (non-private) | 46.2 | 23.7 | 39.4 |
| **Per-device clipping** | | | |
| GPT-3 LoRA ($\epsilon = 0.25$) | 42.0 | 20.4 | 35.7 |
| GPT-3 LoRA ($\epsilon = 1$) | 48.0 | 25.4 | 41.3 |
| GPT-3 LoRA ($\epsilon = 4$) | 48.5 | 26.4 | 42.0 |
| GPT-3 LoRA (non-private) | 53.8 | 29.8 | 45.9 |
| GPT-3 0-shot | 27.4 | 9.0 | 20.9 |
| GPT-3 4-shot | 42.1 | 19.6 | 34.2 |

## 6 RELATED WORK

Training large deep learning models with DP has gained momentum in the recent years. For instance, Anil et al. (2021) privately pretrained BERT models, and Kurakin et al. (2022) privately trained deep ResNets on ImageNet. Recent works have also investigated private fine-tuning (Kerrigan et al., 2020; Tian et al., 2021; Senge et al., 2021; Hoory et al., 2021; Basu et al., 2021; Yu et al., 2021b) and observed that one can achieve favourable privacy-utility trade-offs with large pretrained models for image classification (Luo et al., 2021; Tramèr & Boneh, 2021; Golatkar et al., 2022; De et al., 2022; Mehta et al., 2022) and tasks in NLP (Yu et al., 2022; Li et al., 2022b;a). Group-wise clipping schemes considered in our work improve the efficiency of DP-SGD and further this line of research by making scaling private learning easier.

Several works considered adjusting the clipping threshold of DP-SGD adaptively during training (Pichapati et al., 2019; Asi et al., 2021). The most related to us is that by Andrew et al. (2019) who set the threshold for flat clipping as privately estimated quantile of gradient norms. They showed that doing so eased hyperparameter tuning without affecting the final model performance. Different from these works, ours considers per-layer clipping, where adapting the clipping threshold plays a more crucial role for obtaining good utility. More discussion on related work is in Appendix I.

## 7 CONCLUSION

We showed that group-wise clipping schemes are effective tools to improve the efficiency of DP-SGD for small- to moderate-scale workflows that run on single accelerators, and to avoid overheads in private distributed pipeline parallel training of models that do not fit on single accelerators. We showed that adaptive clipping algorithms can mitigate known utility losses associated with using fixed and hand-tuned thresholds. Designing group-wise clipping algorithms that can Pareto-dominate flat clipping in terms of privacy vs utility (or show impossibility) is an interesting future direction.

---

[5]We believe the chance of contamination occurring for this task to be small. See Appendix C for discussions.

## LIMITATIONS

Group-wise clipping schemes offer various advantages but are not without limitations and drawbacks. First, group-wise clipping algorithms tend to have a few extra hyperparameters. This could lead to a need of additional tuning when optimal hyperparameters differ across tasks and domains (although we showed that across the tasks we studied, the optimal values for most of the additional hyperparameters remained stable). Second, the per-layer clipping scheme gives limited efficiency improvements in non-distributed settings when only few parameters are fine-tuned. Lastly, care must be taken during implementation to fully realize the gains of adaptive per-layer clipping in practice.

## ETHICS STATEMENT

Our work studies and improves differentially private learning algorithms along two distinct axes and has the potential to expand the scope of machine learning on sensitive data. We argue that improvements in differentially private machine learning alone should not be the sole motivation to expand the collection of user data or make aggressive the training of machine learning models on such data without considering the potential long-term harms of developing and releasing models trained with sensitive data.

Our efforts on scaling differentially private fine-tuning to work with GPT-3 are purely motivated by an academic research question. We note there are privacy concerns associated with the pretraining corpus of GPT-3, and thus a model fine-tuned from GPT-3 should not be deployed without undergoing careful privacy audits. For deployment purposes, we suggest fine-tuning only models pretrained on carefully curated corpora.

Lastly, we note that language is inherently complex, and its complexity may well be reflected in datasets for sophisticated tasks such as dialog completion. Differential privacy as a guarantee alone may fail to fulfill the desired privacy goals if example boundaries are not set appropriately.

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

# A    EXPERIMENT DETAILS

## A.1    SETUP FOR CIFAR-10 CLASSIFICATION WITH WRN16-4 MODEL

In the paper, we have used CIFAR-10 classification task with wide residual network (WRN16-4) to demonstrate some points. To increase the reproducibility, we describe the detailed setting of these experiments.

We modify the standard WRN16-4 (Zagoruyko & Komodakis, 2016) following the suggestions of De et al. (2022), i.e., replacing batch normalization with group normalization and using weight standardization for convoluntional weights, except that we do not use augmentation multiplicity for simplicity. Specifically, for flat clipping, De et al. (2022) uses a handed tuned clipping threshold $C = 1$ and learning rate lr $= 4$. Due to the fact that the learning rate and clipping thresholds jointly affect the performance in a complex way, we set the fixed per-layer clipping thresholds and the adaptive per-layer clipping thresholds so that they both have equivalent global threshold $C = 1$. For fixed per-layer clipping, each layer clipping threshold is $C/\sqrt{K}$ where $K$ is the number of layers. For adaptive clipping thresholds $C_1, ..., C_K$, we rescale them $\tilde{C}_k := C \cdot C_k/\sqrt{\sum_k C_k^2}$ for all $k \in [K]$.

To make the comparison more fair, we carefully tune the hyper-parameters of fixed per-layer clipping in Table 1, Table 11 and Figure 3. We find that small fixed thresholds can improve the performance of fixed per-layer clipping. We try different $C$'s from $\{1.0, 0.5, 0.1, 0.05\}$ while making $C \cdot$ lr constant (which is critical for SGD), and set clipping threshold $C_k = C/\sqrt{K}$ for each layer. Finally, for fixed per-layer clipping, We choose the best hyper-parameter combinations ($C = 0.05$, lr $= 40$) and ($C = 0.1$, lr $= 20$) for $\epsilon = \{3, 8\}$, respectively.

For both fixed and adaptive per-layer clippings, we use the global strategy for noise allocation, i.e., $\gamma_k = 1$ for all $k \in [K]$. Moreover, we use the same optimizer, weight decay, momentum, learning rate schedule, batch size and max epochs as flat clipping, as shown in Table 7. We tune the learning rate from two choices $\{2, 4\}$ for all three algorithms. For adaptive per-layer clipping, we use a fraction $r = 0.01$ of privacy budget to estimate quantiles and quantile learning rate $\eta = 0.3$. We tune the target quantile from three choices $\{0.5, 0.6, 0.7\}$. We will evaluate the hyperparameter sensitivity in ablation study (Section F). Hyperparameters are tuned by training from scratch on training set and evaluating on test set. We use the best hyperparameter combinations for different $\epsilon$ respectively and report the test set accuracy of the last epoch in Table 2.

Table 7: Hyper-parameters of flat clipping and per-layer clipping for WRN16-4 on CIFAR-10.

|  | Adaptive per-layer | Fixed per-layer | Flat clipping |
|---|---|---|---|
| **Optimizer** |  | SGD |  |
| **Weight Decay** |  | 0 |  |
| **Momentum** |  | 0 |  |
| **Learning Rate Schedule** |  | Constant |  |
| **Batch Size** |  | 4096 |  |
| **Max Epochs** |  | 300 |  |
| **Learning Rate** | $\{2, 4\}$ | equivalent $\{2, 4\}$ | $\{2, 4\}$ |
| **Allocation Method** | Global | Global | - |
| **Private Quantile Relative Budget** $r$ | 1% | - | - |
| **Quantile Learning Rate** $\eta$ | 0.3 | - | - |
| **Target Quantile** $q$ | $\{0.5, 0.6, 0.7\}$ | - | - |

## A.2    SETUP FOR GLUE TASKS WITH ROBERTA MODELS

To evaluate the performance of adaptive per-layer clipping, we conduct experiments on GLUE tasks by fine-tuning RoBERTa-base and RoBERTa-large models with differential privacy.

The optimizer setup and dropout rates are the same for adaptive per-layer clipping, fixed per-layer clipping, fixed flat clipping and adaptive flat clipping, as shown in Table 8.

For per-layer clipping, we use the global strategy for noise allocation, i.e., $\gamma_k = 1$ for all $k \in [K]$.

We tune other hyperparameters: peak learning rate, batch size, clipping thresholds for fixed per-layer clipping, target quantile $q$ for adaptive per-layer clipping, as shown in the bottom half of Table 8.

To tune hyperparameters fairly, we split the training set of SST-2 into two parts: a new training set containing 80% of original training set and a validation set containing the remaining. We select the best hyperparameters with the performance on the validation set, averaging over 3 different seeds. Table 8 shows the best hyperparameter combinations we use for adaptive and fixed per-layer clipping.

For experiments in Section 5.2, we set the privacy budget for quantile estimation $r = 10\%$. Figure 6 suggests that using smaller values such as $r = 1\%$ or $r = 5\%$ may produce slightly better results.

We transfer hyperparameters tuned on SST-2 to the remaining GLUE tasks. Specifically, we follow Li et al. (2022b) and keep the sampling rate the same across different datasets.

For the GLUE tasks considered, we find that training for more epochs generally improves the performance for both flat and per-layer clipping. To ensure runs finish under realistic training times, we fix the max epochs to be 20 for experiments for the adaptive per-layer clipping runs reported in Table 3. We report the accuracies on the original dev sets for each GLUE task.

For the second experiment in Section 5.2, we only optimize with respect to the self-attention layers and the classification head parameters for both flat clipping and adaptive per-layer clipping so that the total computational cost of this controlled ablation study is manageable.

Table 8: Hyper-parameters of per-layer clipping for RoBERTa on SST-2 dataset, where the **text in bold** denotes the hyper-parameters we eventually use.

| Model | RoBERTa-base | | RoBERTa-large | |
|---|---|---|---|---|
| Method | Adaptive | Fixed | Adaptive | Fixed |
| Optimizer | | Adam | | |
| Adam $(\beta_1, \beta_2)$ | | (0.9, 0.98) | | |
| Adam $\epsilon$ | | $10^{-6}$ | | |
| Weight Decay | | 0 | | |
| Warm-up Ratio | | 0.06 | | |
| Learning Rate Schedule | | Linear Decay | | |
| Dropout | | 0.1 | | |
| Attention Dropout | | 0.1 | | |
| Max Epochs | | 20 | | |
| Peak Learning Rate | $\{1, \mathbf{2}, 4\} \times 10^{-4}$ | $\{\mathbf{1}, 2, 4\} \times 10^{-4}$ | $\{\mathbf{1}, \mathbf{2}, 4\} \times 10^{-4}$ | $\{\mathbf{1}, \mathbf{2}, 4\} \times 10^{-4}$ |
| Batch Size | $\{1, \mathbf{2}, 4\} \times 2^9$ | $\{\mathbf{1}, 2, 4\} \times 2^9$ | $\{\mathbf{1}, 2, 4\} \times 2^9$ | $\{\mathbf{1}, 2, 4\} \times 2^9$ |
| Allocation Method | | Global | | |
| Init Threshold | 1.0 | $\{\mathbf{0.1}, 0.5, 1.0\}$ | 1.0 | $\{0.1, 0.5, \mathbf{1.0}\}$ |
| Private Quantile Relative Budget $r$ | 10% | - | 10% | - |
| Quantile Learning Rate $\eta$ | 0.3 | - | 0.3 | - |
| Target Quantile $q$ | $\{0.5, 0.75, \mathbf{0.85}\}$ | - | $\{0.5, 0.75, \mathbf{0.85}\}$ | - |

### A.3 SETUP FOR LANGUAGE GENERATION TASKS WITH GPT-2

We reused most of the hyperparameters specific to adaptive quantile estimation based on tuning results on SST-2. We retuned the target quantile parameter as we observed that optimal values of this parameter tend to be different for different tasks. To ensure a fair comparison against full fine-tuning, we constrain the runs with adaptive per-layer clipping to have the same batch size and training epochs as in (Li et al., 2022b). We adopted the default values set by the Hugging Face `transformers` library for Adam's $\beta_1$, $\beta_2$, and $\epsilon$. Table 9 contains the full set of hyperparameters.

## B ADDITIONAL EXPERIMENTS ON GRADIENT NORM SHIFT

In this section, we illustrate the distribution of gradient norms shift in both CIFAR-10 training and SST-2 fine-tuning. To visualize the gradient norms, we first randomly select some samples from the training set, and take the checkpoints at different epochs of a privately trained model with adaptive per-layer clipping and the privacy parameter $\epsilon = 8$. For each sample, we compute the gradient norm of each layer. Specifically, for CIFAR-10, we ramdonly select 32 samples and place layers of WRN16-4 from input (left) to output (right) in Figure 2.

For SST-2, we randomly select 4,096 samples and some layers in the RoBERTa-base model, and plot the histogram of the per-sample per-layer gradient norms in Figure 4 across the first few epochs. It is

Table 9: Hyperparameters for full fine-tuning GPT-2 with adaptive per-layer clipping. Numbers in bold are best performing hyperparameters used for reporting final results.

| Model | GPT-2 | |
|---|---|---|
| Dataset | E2E | DART |
| Optimizer | Adam | |
| Adam ($\beta_1$, $\beta_2$) | (0.9, 0.999) | |
| Adam ($\epsilon$) | 1e-8 | |
| Weight Decay | 0 | |
| Learning Rate Schedule | Linear Decay | |
| Batch Size | 1000 | 1500 |
| Max Epochs | 10 | 15 |
| Learning Rate | $2 \times 10^{-3}$ | |
| Allocation Method | Global | |
| Init Threshold | 0.01 | |
| Private Quantile Relative Budget $r$ | 1% | |
| Quantile Learning Rate $\eta$ | 0.3 | |
| Target Quantile $q$ | $\{\mathbf{0.3}, 0.5, 0.7, 0.9\}$ | reuse best of left |

worth noting that we found that for fine-tuning SST-2 with RoBERTa-base, it is true for many layers that the 85% clipping threshold (see red dashed line in Figure 4) is just the point can split samples into a group with small gradient norms and a group with large ones.

Both of Figure 2 and Figure 4 demonstrate that the distribution of gradient norms is complex and may related to many factors: (1) **Iterations & Samples**: gradient norms are small and spread out across layers in the early epochs, and as the training process goes on, per-sample gradients become divided, the large becomes larger and the small becomes smaller; (2) **Layers**: gradient norms of layers close to the input are larger than those of layers close to output, it is more prominent in the later stages of training, but it's aligned well across samples.

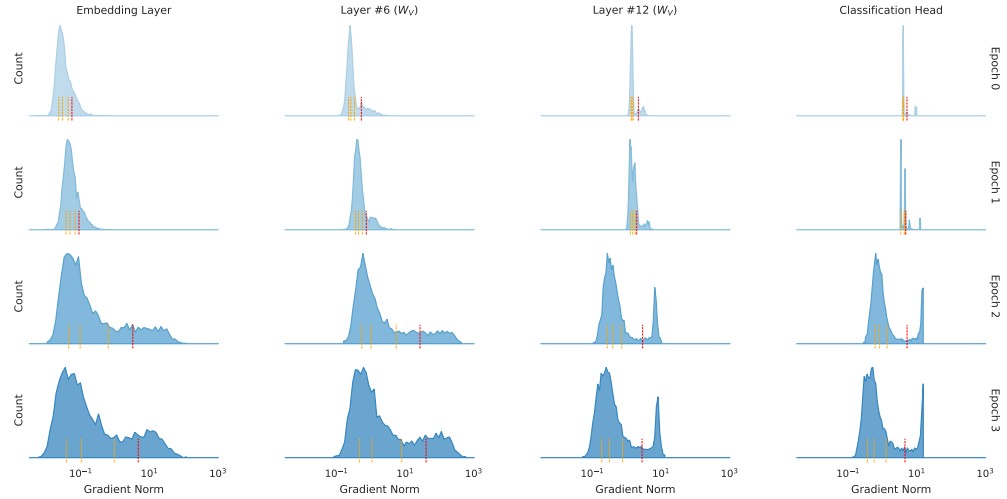

Figure 4: The distribution of gradient norms of 4,096 randomly selected samples shifts across the fine-tuning procedure for the setup of SST-2 with RoBERTa-base. We show the gradient statistics of (1) the embedding layer, (2) the value matrix $W_V$ of the 6th and 12th self-attention module, (3) the final classification layer. Quantiles $\{25\%, 50\% \text{ (median)}, 75\%, 85\%\}$ of the gradient norms are marked with dashed lines, from left to right.

## C  MORE ON PER-DEVICE CLIPPING AND EXPERIMENTS WITH GPT-3

### C.1  ADDITIONAL COMMENTS ON PER-DEVICE CLIPPING

Algorithm 2 details the private pipeline parallel training procedure with per-device clipping covered in the main text. The algorithm adopts the equal budget noise allocation strategy to avoid incurring extra communication across devices. For simplicity, the pseudocode covers a single update and omits any subprocedures for adapting clipping thresholds. Extending this pseudocode to adaptive threshold clipping based on quantile estimation is straightforward. Lastly, the pseudocode assumes that the model has already been partitioned into chunks of consecutive layers, where each chunk $\boldsymbol{\theta}_k$ is hosted on device $k$.

---

**Algorithm 2** Single Update of Private Pipeline Parallel Training With Per-Device Clipping

---

1: **INPUT**: Minibatch $\mathcal{S}$; iterate $\boldsymbol{\theta}$; per-device parameters $\{\boldsymbol{\theta}_1, \cdots, \boldsymbol{\theta}_K\}$ hosted on $K$ devices; clipping thresholds $\{C_1, \ldots, C_K\}$; noise multiplier $\sigma$; learning rate $\eta$; number of microbatches per minibatch $J$
2: Partition minibatch $\mathcal{S}$ into $J$ microbatches $\{\mathcal{S}_1, \cdots, \mathcal{S}_J\}$
3: Create an empty execution schedule $\mathcal{C}$
4: **for** $j = 1$ to $J$ **do**
5:     Stage microbatch $\mathcal{S}_j$'s `LocalForward` and `LocalBackward` calls in the schedule $\mathcal{C}$, ensuring the stages are executed sequentially for this microbatch
6: **end for**
7: Organize the schedule $\mathcal{C}$ based on pipeline parallel rules, allowing different devices to process different microbatches simultaneously
8: Execute the schedule $\mathcal{C}$
9: Synchronize all devices
10: **for** $k = 1$ to $K$ **do**
11:     $\boldsymbol{\theta}'_k \leftarrow \boldsymbol{\theta}_k - \eta u_k$.
12: **end for**
13: **return** $\boldsymbol{\theta}' = (\boldsymbol{\theta}'_1, \cdots, \boldsymbol{\theta}'_K)$

---

**Algorithm 3** `LocalForward`

---

1: **INPUT**: Device id $k$; microbatch index $j$
2: Wait for activations $a_{k-1}^{(j)}$ from device $k-1$ if $k > 1$; otherwise transfer microbatch $\mathcal{S}_j$ onto device $k$
3: Perform forward pass with $a_{k-1}^{(j)}$ and model piece $\boldsymbol{\theta}_i$ (stored on device $k$) to obtain outputs $a_k^{(j)}$
4: Store a copy of $a_k^{(j)}$ on CPU if not already saved
5: Communicate activations $a_k^{(j)}$ to device $k + 1$ if $k < K$

---

**Algorithm 4** `LocalBackward`

---

1: **INPUT**: Device id $k$; microbatch index $j$
2: Transfer activations $a_{k-1}^{(j)}$ from CPU to device $k$
3: Wait for output gradients $o_k^{(j)}$ if device $k < K$
4: Rematerialize activations by performing extra forward pass
5: Clip and sum per-example gradients $\{\bar{g}_k^{(i)}\}_i$ of local model piece $\boldsymbol{\theta}_k$ by back-propagating based on $o_k^{(j)}$ or the loss values with threshold $C_k$; add this to a local accumulator $u_k$ (stored on device $k$)
6: If $j = 1$, add noise to accumulator $u_k$ to guarantee DP
7: Compute gradients with respect to input $o_{k-1}^{(j)}$ and communicate this to device $k-1$ if $k > 1$.

---

### C.2  FINE-TUNING GPT-3 ON SAMSUM

Note the term "GPT-3" in the literature is used in multiple occasions and can refer to multiple models. Our experiments are based on fine-tuning or prompting the original GPT-3 model (Brown et al.,

2020) and not the more recent variants which had been fine-tuned or adapted in some way (e.g., instruct-GPT-3 (Ouyang et al., 2022) labeled with prefix `instruct-` in OpenAI API).

Larger models are known to have better fine-tuned performance when inputs and outputs are formatted as instructions and responses (Wei et al., 2021; Sanh et al., 2021). We observed similar results when fine-tuning with differential privacy, and thus augmented the training and test sets by prepending the inputs with the instruction "Summarize the following dialogue" and the outputs with the delimiter "TL;DR". To ensure a fair comparison, we used this instruction-augmented dataset for all experiments.

For decoding from models, we used beam search with a beam size of 4 for both GPT-3 and GPT-2-xl (including in-context learning experiments).

To ensure we account for the variability in performance with different prompts for in-context learning, we sampled 3 sets of prompts for the 4-shot learning experiments and reported the average metric over runs.

Without access to the original pretraining corpus, we cannot completely rule out the possibility of data contamination, which refers to the unfortunate outcome that parts of the fine-tuning or evaluation data occur in the pretraining corpus. Nevertheless, we believe the chances of this happening are small due to two reasons. First, zero-shot prompting GPT-3 with both low temperature sampling and beam search based on instruction-augmented inputs tended to result in completions which either repeated or extended the instruction or the dialog (e.g., "the following is a dialog between..."), or attempted to continue the dialog but digressed. In the limited number of examples we inspected, we were unable to find an instance where the output looked similar to a high-quality summary. Second, we looked up the initial time when the SAMSum paper was released to arXiv (late Nov. 2019). Given that the GPT-3 model we based our experiments off were pretrained with shards of Common Crawl uploaded (possibly) at the end of 2019 (Brown et al., 2020), we performed simple searches of the SAMSum paper with their url index in the Dec. 2019 crawl archive of Common Crawl and were not able to find the link of the paper. Notably the SAMSum dataset was crafted by linguists and highly curated (as opposed to collected based on web data).

For fine-tuning GPT-3 with DP LoRA on SAMSum, we reused hyperparameters adopted by Hu et al. (2021), but re-tuned the learning rate based on preliminary runs for another dataset. We set all per-device clipping threshold to be 1e-5 and adopted the equal budget noise allocation strategy for simplicity. We fine-tuned for 5 epochs in all runs (both GPT-3 and GPT-2-xl; both private and non-private).

For the DP LoRA fine-tuning runs, we used a machine with 16 V100 GPUs each with 32 gigabytes of VRAM. This enabled LoRA fine-tuning with a rank of 32 with a microbatch size of 1 under pipeline parallelism. Fine-tuning with DP LoRA for 5 epochs on SAMSum's training set took 15 hours, and decoding with test inputs using beam search further took another 22 hours.

## D  PROOFS

We present the proof for Proposition 3.1. For easy reference, we restate the proposition here.

**Proposition 3.1.** *Let $\sigma$ be the original noise multiplier for noising parameter updates to achieve a certain level of differential privacy (without private quantile estimation) and $\sigma_b$ be chosen for noising quantile estimates (release of the latter consumes $r$ fraction of the privacy budget). Then the new noise multiplier $\sigma_{new}$ for noising parameter updates (consuming $1 - r$ fraction of the budget) is*

$$\sigma_{new} = (\sigma^{-2} - K/(2\sigma_b)^2)^{-1/2}. \tag{3.1}$$

*Proof.* The proof is based on direct calculation, a simple version of which is given in Andrew et al. (2019). First, we note that the clip counts $b_k^{(i)}$ is either 0 or 1 (see line 10 in Algorithm 1). One can make it to be symmetric by using $b_k^{(i)} - \frac{1}{2}$, whose sensitivity is $\frac{1}{2}$. Suppose the gradient has sensitivity $S$.

For the Gaussian mechanism, to keep the privacy budget constant we have

$$S^2/(S\sigma)^2 = S^2/(S\sigma_{\text{new}})^2 + K \cdot \frac{(1/2)^2}{\sigma_b^2}.$$

Simplifying the above expression, we get $\sigma_{\text{new}}$.

We can further compute the fraction $r$ of budget that is used to privately estimate quantiles by $r = K \cdot \frac{(1/2)^2}{\sigma_b^2}/(1/\sigma^2)$. We can also derive the value of $\sigma_b$ given $r$ from the above formula. □

## E  NOISE ALLOCATION COMPARISON

We compare the noise allocation strategies empirically. Apart from the *global strategy* where $\gamma_k = 1$ for all $k \in [K]$ and the *equal budget strategy* where $\gamma_k = C_k$ for all $k \in [K]$ that are discussed in Section 3.3, we also consider another *weighted strategy*: $\gamma_k = C_k/\sqrt{d_k}$ for $k \in [K]$. In this case, the number of parameter plays a role so that each coordinate would roughly have the same signal to noise ratio and the total noise has squared $\ell_2$ norm $V_E \propto (\sum_{k=1}^K d_k) \cdot (\sum_{k=1}^K C_k^2)$.

We fine-tune RoBERTa-base models on the SST-2 sentence classification task. The hyper-parameters are searched for each strategy separately where the ranges follow Appendix A. Results are presented in Table 10. We can see that three strategies achieves comparable performance and the global strategy is slightly better. Therefore, we use global strategy for all the experiment except for GPT-3 where the equal budget strategy is used to eliminate the concern of communication across devices.

Table 10: Accuracy (in %) on SST-2 dev set with different noise allocation strategies.

| Model | Strategy | $\epsilon = 3$ | | $\epsilon = 8$ | |
|---|---|---|---|---|---|
| | | Training | Validation | Training | Validation |
| RoBERTa-base | Global | **89.2** | **92.0** | **89.7** | **92.3** |
| | Equal Budget | 88.3 | 91.4 | 89.0 | 91.7 |
| | Weighted (equal SNR) | 89.0 | 91.7 | 89.6 | 91.9 |

## F  ABLATION STUDIES

Here we conduct ablation studies to see 1) the influence of using different quantiles to perform clipping; 2) the influence of varying the privacy budget for quantile estimation; 3) whether adaptive flat clipping significantly better than fixed flat clipping.

**Clipping with Different Target Quantiles.** We use different target quantiles for clipping on both WRN16-4 and RoBERTa-base. We choose the quantile for CIFAR-10 from $\{0.3, 0.4, 0.5, 0.6, 0.7, 0.8, 0.9\}$ and that for SST-2 from $\{0.05, 0.2, 0.4, 0.6, 0.85, 0.9, 0.95\}$. Other hyperparameters are the same as those in Section 5.1 and 5.2. We plot the results in Figure 5. On CIFAR-10, the accuracy is robust to the choice of target quantile on all values considered. On SST-2, all quantiles around 0.9 give good performance. This suggests that setting the target quantile according to the model accuracy is a good default choice for the classification tasks. For generation task, we tune the target quantile as a hyper-parameter in general.

**Different Budgets for Quantile Estimation.** We show the influence of using different privacy budgets to estimate the target quantile. We fine-tune RoBERTa-base models on SST-2. The fraction of privacy budget for quantile estimation $r$ is from $\{0.01\%, 0.1\%, 1\%, 5\%, 10\%, 20\%, 40\%, 80\%\}$. We plots the results in Figure 6. The performance is good for a wide range of $r$. When $\epsilon = 8$, using $r$ as small as $0.01\%$ still gives good accuracy. This further confirms the finding in Andrew et al. (2019) that quantiles can be estimated quite accurately with small privacy budget. Therefore we only need to split negligible budget for the private quantile estimation without affecting much the noises added to the model updates.

**Adaptive Per-layer Clipping vs Adaptive Flat Clipping.** We have verified that adaptive per-layer clipping can match the performance of well-tuned flat clipping in Section 5. To really justify value of adaptive per-layer clipping, we need to demonstrate that adaptive flat clipping does not achieve significantly better performance than fixed flat clipping. We run experiments on the CIFAR-10 task with WRN16-4 and the SST-2 task with RoBERTa-base model. Their results are presented in

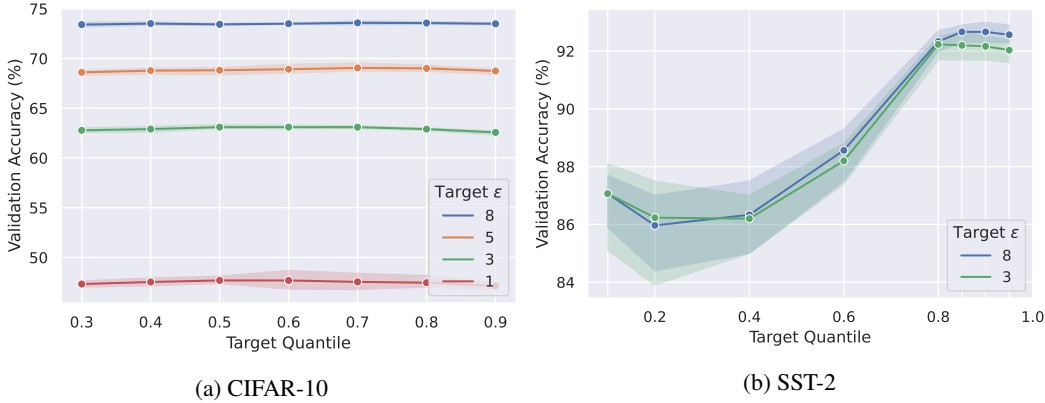

Figure 5: Validation accuracy (in %) with different target quantiles.

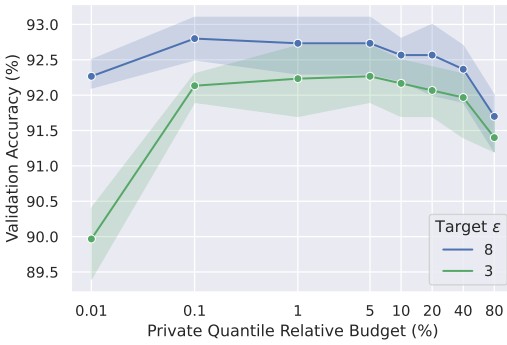

Figure 6: Validation accuracy (in %) on SST-2 of using different budgets for quantile estimation.

Table 11a and Table 11b. We can see that adaptivity also helps flat clipping but the improvement is not statistically significant. The performance of adaptive per-layer clipping is on par with that of adaptive flat clipping as well.

Table 11: Adaptivity helps flat clipping but not as much as for per-layer clipping. Averaged accuracy and standard deviation are given by 3 independent runs.

(a) CIFAR-10

| Method | $\epsilon = 3$ | $\epsilon = 8$ |
|---|---|---|
| **Flat clipping** | | |
| fixed | $63.1_{(0.22)}$ | $73.9_{(0.87)}$ |
| adaptive | - | - |
| | +0 | +0 |
| **Per-layer clipping** | | |
| fixed | $60.6_{(0.79)}$ | $67.8_{(1.20)}$ |
| adaptive | $63.7_{(0.34)}$ | $73.5_{(0.87)}$ |
| | +3.1 | +5.7 |

(b) SST-2

| Method | $\epsilon = 3$ | $\epsilon = 8$ |
|---|---|---|
| **Flat clipping** | | |
| fixed | $91.5_{(1.07)}$ | $92.0_{(0.67)}$ |
| adaptive | $92.2_{(0.70)}$ | $92.6_{(0.46)}$ |
| | +0.7 | +0.6 |
| **Per-layer clipping** | | |
| fixed | $89.4_{(1.04)}$ | $89.7_{(0.70)}$ |
| adaptive | $92.0_{(0.23)}$ | $92.4_{(0.44)}$ |
| | +2.6 | +2.7 |

## G    ADDITIONAL EXPERIMENTS ON RUN TIME

We include additional experiments comparing the run time of different clipping approaches. Our goal is to further consolidate the claims that 1) adaptive per-layer clipping attains similar or better task performances under the same epoch budget for certain workflows, and that 2) this translates into compute time savings since per-layer clipping is faster than alternative clipping approaches per update (or almost equivalently per epoch). All experiments here are performed on a machine with a single Titan RTX GPU with 24 GB of VRAM (different from the configuration in Figure 1 which uses a single A6000 GPU).

The direct experiment we perform is to full fine-tune GPT-2 on E2E with three clipping approaches (adaptive per-layer, ghost, and flat clipping) under the same epoch constraint (which we fix to be 10 for all workflows). Regarding hyperparameters for flat clipping, we adopt the set of values obtained from extensive tuning on this task used by Li et al. (2022b). We reuse the same set of hyperparameters values for ghost clipping, since the approach essentially results in the same gradient updates as flat clipping up to numerical precision (only computed in a different way). Using these near optimal hyperparameters for flat and ghost clipping prevents our experiments from unfairly disfavouring the two approaches. Figure 7 shows that adaptive per-layer clipping consistently achieves lower test set negative log-likelihood than flat clipping and ghost clipping under any given wall time elapse. While language generation metrics (e.g., BLEU and ROUGE-L) are generally noiser than the test set NLL, Figure 8 shows that adaptive per-layer clipping generally yields better task metric numbers compared to flat clipping and ghost clipping under the same wall time.

Finally, we note the caveat that the precise run time advantage of adaptive per-layer clipping against flat clipping may vary across machines and GPU types. In addition, the realized gains for actual training workflows might be smaller than that observed in our controlled experiments (e.g., Figure 1) due to compute time spent on auxiliary operations such as data loading and data preprocessing (e.g., pad sequences of different length to the same length). For instance, we repeat the controlled experiment in Figure 1 but this time with a different GPU, and observe slightly different factors of speed gains. Overall, we generally see that adaptive per-layer clipping is above 1.4x the speed of flat clipping in our controlled experiments, and the realized gains is roughly as much for full fine-tuning on E2E with our implementation.

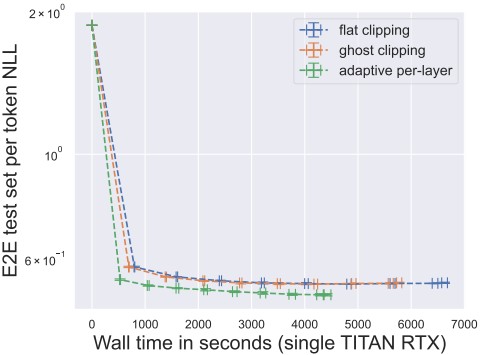

Figure 7: Adaptive per-layer clipping consistently achieves lower test set negative log-likelihood than flat clipping and ghost clipping under the same wall time.

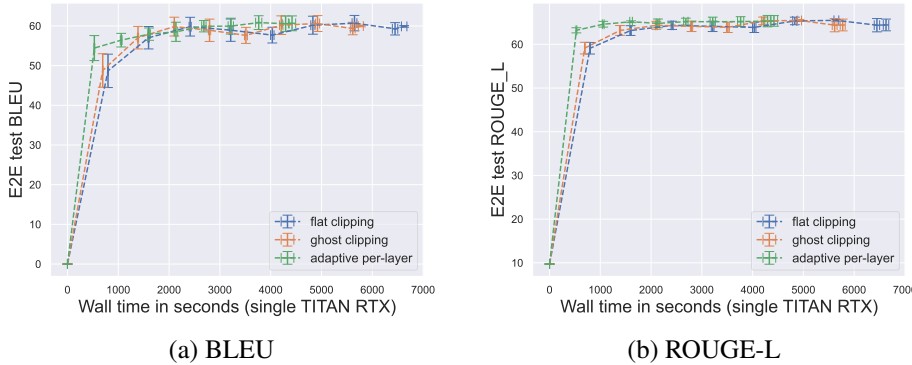

(a) BLEU  (b) ROUGE-L

Figure 8: Adaptive per-layer clipping generally yields better task metric numbers compared to flat clipping and ghost clipping under the same wall time.

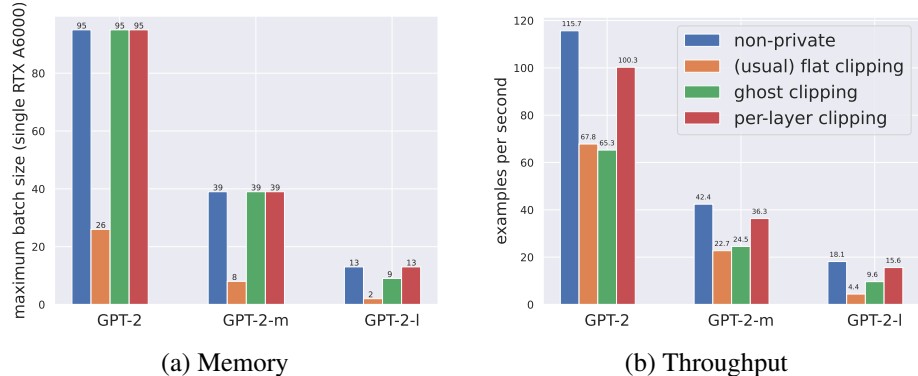

(a) Memory  (b) Throughput

Figure 9: Private learning with (adaptive) per-layer clipping can be almost as efficient as non-private learning (the throughput gap is less than 15% in this case).

# H  ADDITIONAL EXPERIMENTS COMPARING ADAPTIVE PER-LAYER AGAINST FLAT CLIPPING

This section complements the results in Table 4 and follows the same experimental protocol for those experiments except now we consider $\epsilon = 8$. The goal is to provide further evidence showing that adaptive per-layer clipping has a privacy-utility tradeoff that is comparable to flat clipping when both methods are used to train models for the same number of training epochs.

Table 12 shows that for full fine-tuning and across the epoch constraints we considered, adaptive per-layer clipping is competitive with flat clipping in terms of accuracy for different privacy budgets. Note the results we obtain for flat clipping in Table 12 are higher than those reported by Li et al. (2022b). This is because to ensure that we are not unfairly disfavouring flat clipping for this SST-2 task, we tuned hyperparameters on this task (details in Appendix A.2); recall Li et al. (2022a) didn't tune on SST-2, but instead relied on hyperparameter transfer from the E2E task.

Table 12: Adaptive per-layer clipping matches or outperforms flat clipping in accuracy (%) on SST-2 under fixed epoch ($E$) constraints. This experiment performs full fine-tuning with both clipping approaches. Numbers in parentheses are standard deviations from three independent runs.

| Model | Method | $\epsilon = 8$ | | | |
|---|---|---|---|---|---|
| | | $E = 3$ | $E = 10$ | $E = 20$ | $E = 30$ |
| RoBERTa-base | Flat clipping (tuned) | 89.13(0.64) | 91.53(0.12) | 91.97(0.67) | 92.10(0.56) |
| | Adaptive per-layer | 90.83(1.39) | 92.27(0.76) | 92.63(0.32) | 92.87(0.12) |
| RoBERTa-large | Flat clipping (tuned) | 92.43(0.32) | 93.50(0.20) | 94.57(0.55) | 94.87(0.76) |
| | Adaptive per-layer | 93.20(0.36) | 93.53(0.47) | 94.37(0.15) | 94.33(0.38) |

# I  ADDITIONAL RELATED WORK

**Faster DP-SGD.**  Improving the efficiency of DP-SGD is an active research area. One line of works improve the implementation without changing the algorithm such as using better parallelism and compile-time optimization (Subramani et al., 2021; Anil et al., 2021). Subramani et al. (2021) show that the running time of carefully implemented DP-SGD is comparable to non-private SGD for small-size models. However, the high memory cost of storing per-example gradients still limits the throughput of DP-SGD when the model size is large (Li et al., 2022b). Another line of works avoids instantiating per-example gradients by running backpropagation twice (Goodfellow, 2015; Lee & Kifer, 2021; Bu et al., 2021; Li et al., 2022b; Bu et al., 2022). The high-level idea is to compute or estimate per-example gradient norms in the first backpropagation and reweight loss functions before the second backpropagation. Although these works achieve memory efficiency, they add computational overhead because of the additional backpropagation.

**DP-SGD with Per-layer Clipping.**  Per-layer clipping (or more generally group-wise clipping) has been studied in Abadi et al. (2016); McMahan et al. (2018a;b); Dupuy et al. (2022). However, the advantage of per-layer clipping has not yet been fully understood because of two reasons. Firstly, previous work does not focus on computational efficiency, leaving the empirical advantage of group-wise clipping unexplored. Secondly, these works simply adopt fixed thresholds for per-layer clipping and hence generally observe performance drops compared to flat clipping. In this work, we use adaptive thresholds to improve the privacy-utility tradeoff of per-layer clipping. Moreover, we give an efficient implementation of per-layer clipping to demonstrate its superior empirical advantage.

**Privacy Attacks Against Deep Models.**  Deep models may unintentionally leak sensitivie information about their training data (Shokri et al., 2017; Hitaj et al., 2017; Zhu et al., 2019; Song et al., 2019; Carlini et al., 2020; Choquette-Choo et al., 2021; Carlini et al., 2022; Balle et al., 2022). For instance, Carlini et al. (2020) show that GPT-2 outputs its training data when short prefixs are provided. Training deep models with differential privacy has become a popular choice to prevent data leakage (Abadi et al., 2016; Papernot et al., 2017; McMahan et al., 2018b; Zhu et al., 2020). In addition to theoretical guarantee, differentially private models are also very robust to empirical privacy attacks (Bernau et al., 2019; Carlini et al., 2019; Yu et al., 2021b).

**Adapting to the Geometry of Gradients in DP-SGD.**  Using adaptive clipping thresholds in DP-SGD fits more broadly into a line of work that adapts the geometry of gradients to clipping and noising. The gradients of machine learning models usually have much smaller intrinsic dimensions than the model sizes. This property has been used to prove better theoretical bounds for DP-SGD or improve its empirical performance (Kairouz et al., 2020; Song et al., 2020; Zhou et al., 2021; Yu et al., 2021a; Li et al., 2022a; Ma et al., 2022).

