# OpenReview forum: "Exploring the Limits of Differentially Private Deep Learning with Group-wise Clipping"
_ICLR.cc/2023/Conference — ICLR 2023 poster_

### Official Review · Reviewer_FnUx · 2022-10-23

**Confidence:** 3
**Correctness:** 4
**Technical Novelty And Significance:** 2
**Empirical Novelty And Significance:** 3
**Recommendation:** 6

**Clarity, Quality, Novelty And Reproducibility:**

Clarity and quality is good, novelty is not strong, reproducibility is fair since code is not provided.

**Strength And Weaknesses:**

The main strength is the successful training of GPT-3 to reasonable performance with relatively small privacy budget, which is achieved by device-wise gradient clipping and parameter tuning.

Weakness is mainly on novelty, the work is just applying existing DP training algorithms to training GPT-3, with some hyperparameter tuning.

**Summary Of The Paper:**

This work conducted experiments on privately fine-tuning GPT-3 using group-wise gradient clipping and successfully trained a model performing well with relatively small privacy budget.

**Summary Of The Review:**

I feel it is interesting to see GPT-3 can perform quite well event with privacy budget as small as 1, but I feel novelty might be a concern for acceptance.

---

> ### Author Response · Authors · 2022-11-18
> **Thank you for your review; our response**
>
> We thank the reviewer for their feedback. Below we address the main concern.
>
> > Weakness is mainly on novelty, the work is just applying existing DP training algorithms to training GPT-3, with some hyperparameter tuning.
>
> We thank the reviewer for appreciating our efforts on scaling DP fine-tuning to GPT-3. However, we believe that the reviewer has overlooked several of our key contributions, thus likely making their evaluation of our work uncalibrated.
>
> **Novelty of scaling DP fine-tuning with per-device clipping**
>
> First of all, GPT-3 175B is a large model that cannot be hosted on a single modern accelerator. This model is more than 100x larger than the largest previously DP fine-tuned model (GPT-2 xl) for NLP tasks [1]. Hence, developing and implementing an algorithm to efficiently privately fine-tune GPT-3 is a significant undertaking that requires algorithmic innovation as well as a clear understanding of distributed training.
>
> The naive solution of using DP-SGD with flat clipping gives both implementation complications and substantial computational overhead (as detailed in our original draft; see section 4 and appendix). Our per-device clipping method, coupled with the equal budget noise allocation strategy (detailed at top of page 6), is a new approach for reducing implementation complexities and the runtime overhead for DP fine-tuning where large-scale models are critical for achieving the best performance possible.
>
> We thus politely disagree with the reviewer that our work is
> “just applying existing DP training algorithms to training GPT-3, with some hyperparameter tuning.”
>
> We have updated the section on per-device (Section 4) to clarify some of the challenges associated with scaling DP fine-tuning for this model.
>
> **Novelty of adaptive per-layer clipping, and its computational and performance implications**
>
> The GPT-3 component of our work is significant, but so is the other major component, which **studies the computational and performance implications of adaptive per-layer clipping (sections 3, 5.1, and 5.2 of the original draft)**. We suspect that the reviewer may have overlooked this component and missed a substantial portion of our contributions.
>
> We’d like to clarify to the reviewer (in the case they missed section 3 of the paper) that the other important research question (stated in section 1, page 1) we study in this work is
> “Can private learning be as memory and time efficient (per-epoch) as non-private learning?”
>
> This research question is both practically important and academically interesting. As stated in section 1, past works have criticized DP training for its computational inefficiency. Some works in the literature even leverage this argument to sidestep DP and adopt/propose heuristic privacy approaches (most of which have not stood against the test of time and are revealed to cause real privacy concerns under modest privacy attacks).
>
> Our argument is that these inefficiencies are not inherent to DP training, and can be (almost) eliminated through our algorithmic technique (adaptive per-layer clipping; see section 3) and careful implementation. In fact, our results show that for certain workflows, DP training can be almost as fast and memory efficient as non-private training per epoch (Figure 1) without sacrificing task performance (section 3.3). Aside from being directly useful, we believe these results also improve our general understanding of the computational aspects of DP training algorithms.
>
> Overall, we thank the reviewer for their optimism in our work. However, we believe the reviewer may have misunderstood and/or overlooked several of our main contributions. We’d like to encourage the reviewer to holistically look at our work and reconsider their assessment.
>
> [1] Yu, Da, Saurabh Naik, Arturs Backurs, Sivakanth Gopi, Huseyin A. Inan, Gautam Kamath, Janardhan Kulkarni et al. "Differentially private fine-tuning of language models." arXiv preprint arXiv:2110.06500 (2021).

---

### Official Review · Reviewer_QMKV · 2022-10-24

**Confidence:** 4
**Correctness:** 4
**Technical Novelty And Significance:** 3
**Empirical Novelty And Significance:** 4
**Recommendation:** 8

**Clarity, Quality, Novelty And Reproducibility:**

The paper is written clearly with good quality. The exploration of using per-layer clipping and the engineering efforts for large model training is novel. Pseudocode and hyper-parameter are given in the paper but reproducing the implementation and the experiments can be hard.


**Strength And Weaknesses:**

Strengths

1. For the very first time, demonstrated how to train DP-SGD on GPT-3 which is already memory consuming for non-private learning. The engineering efforts that made DP-SGD training working on such a large model are incredible.
2. The paper is well written and the methods are clearly explained. The idea of using per-layer clipping to improve efficiency is novel and 3. not explored in prior works. The intuition of using adaptive clipping to improve the utility is also well-explained.
3. The experiments are comprehensive and convincing. Multiple baselines are used to compare with the proposed methods.

Weaknesses

Not sure how easy it is to reproduce the implementation of the per-layer clipping and parallel training of the large model. There is no discussion on open sourcing but I hope the author would share their implementation for the benefit of the research community.


**Summary Of The Paper:**

This paper explored fine-tuning extremely large models with DP-SGD and proposed methods to reduce the memory and improve the efficiency of the per-example gradient clipping. The authors demonstrated that per-layer gradient clipping with adaptively tuned clipping bounds can be as efficient as non-private training and achieve similar utility to flat clipping. The methods were compared with prior works and evaluated on multiple standard benchmark datasets and models, and for the first time, on the GPT-3 model.


**Summary Of The Review:**

I support accepting this paper given its strong empirical contribution.

---

> ### Author Response · Authors · 2022-11-18
> **Thank you for your detailed feedback and comments; our response**
>
> We thank the reviewer for their detailed feedback and comments. Below we address their concern.
>
> > Not sure how easy it is to reproduce the implementation of the per-layer clipping and parallel training of the large model. There is no discussion on open sourcing but I hope the author would share their implementation for the benefit of the research community.
>
> We will release all source code for reproducing the GLUE experiments, the GPT-2 fine-tuning experiments, and the CIFAR-10 experiments. We’d love to release the implementation for fine-tuning the GPT-3 model, but due to policy issues, we are unable to do so.
>
> Nevertheless, we will make our best effort to provide guidelines for a re-implementation and document potential pitfalls to ease researchers who would like to reproduce our results with a publicly available model.

---

### Official Review · Reviewer_CEDx · 2022-10-25

**Confidence:** 2
**Clarity, Quality, Novelty And Reproducibility:** The proposed work is novel and the pa…
**Correctness:** 3
**Technical Novelty And Significance:** 3
**Empirical Novelty And Significance:** 3
**Recommendation:** 8

**Strength And Weaknesses:**

Strength:
1. The propose work is addressing a fundamental problem in privacy-aware large scale deep learning models in NLP.
2. The proposed method is easy to follow and very intuitive, and quite reasonable.
3. The experiments are pretty solid.

Weakness:
1. The efficiency of the proposed strategy is mentioned in introduction but not evaluated in experiments.
2. The epsilon value is set in relatively large values (3 and 8) in several experiments.

**Summary Of The Paper:**

The paper proposes a novel adaptive per-layer clipping and shows superior performance compare with classical flat clipping in GPT-2 and GPT-3 to protect the privacy while maintain a better task performance.

**Summary Of The Review:**

The paper proposes a novel adaptive per-layer clipping and shows superior performance compare with classical flat clipping in GPT-2 and GPT-3 to protect the privacy while maintain a better task performance.

Strength:
1. The propose work is addressing a fundamental problem in privacy-aware large scale deep learning models in NLP.
2. The proposed method is easy to follow and very intuitive, and quite reasonable.
3. The experiments are pretty solid.

Weakness:
1. The efficiency of the proposed strategy is mentioned in the introduction but not evaluated in experiments.
2. The epsilon value is set in relatively large values (3 and 8) in table 3 and table 4 without explanation.

I recommend an acceptance.

---

> ### Author Response · Authors · 2022-11-18
> **Thank you for your review; our response**
>
> We thank the reviewer for their feedback and positive response. Below we address the main concerns.
>
> > The efficiency of the proposed strategy is mentioned in introduction but not evaluated in experiments.
>
> We thank the reviewer for raising this point. Regarding the efficiency of adaptive per-layer clipping, **our original draft contains a profiling experiment whose results are detailed in Figure 1**. Results show that private learning with adaptive per-layer clipping approaches the efficiency of non-private learning (roughly 15% overhead) for this particular problem. In addition, adaptive per-layer clipping has substantially higher throughput than flat and ghost clipping.
>
> To additionally clarify the point of efficiency, **we have included additional experiments in Appendix G** showing that adaptive per-layer clipping, compared to fixed flat clipping, reaches a given target task performance (with a fixed epoch budget) under less wall time, using language generation problems as the main testbed. Our experiments show that adaptive per-layer clipping is >1.4x as fast as flat clipping per epoch and generally attains better test set NLL, BLEU, and ROUGE-L scores than flat clipping and ghost clipping under fixed wall times.
>
> > The epsilon value is set in relatively large values (3 and 8) in several experiments.
>
> We agree with the reviewer that setting epsilon to 3 and 8 gives less strong a privacy guarantee compared to setting epsilon to 1.
>
> However, we argue that privacy budgets like 3 and 8 are commonly used in the literature in past works. For instance, [1] considers epsilons among 2, 4, 8, [2] considers epsilon values 2, 5, 8, [3] considers epsilon values ranging from 1 to 3, and [4] considers epsilon values 3 and 8. Note the above works are all considered influential in this space, e.g., [1] has more than 3k citations, [2] won a best paper award at ICLR 2017, [3] was selected for spotlight presentation at ICLR 2021, and [4] was selected for oral presentation at ICLR 2022. Furthermore, setting epsilon to 3 and 8 enables a fair comparison of our work against prior art and facilitates future work to compare against ours so that this empirical research field can move forward.
>
> In addition, recent works also provide evidence that DP-trained models are empirically resistant against data reconstruction attacks even with relatively large values of epsilon [5, 6, 7].
>
> Lastly, we have included clarifications in the main text to address this concern.
>
> [1] Abadi, Martin, Andy Chu, Ian Goodfellow, H. Brendan McMahan, Ilya Mironov, Kunal Talwar, and Li Zhang. "Deep learning with differential privacy." In *Proceedings of the 2016 ACM SIGSAC conference on computer and communications security* , pp. 308-318. 2016.
>
> [2] Papernot, Nicolas, Martín Abadi, Ulfar Erlingsson, Ian Goodfellow, and Kunal Talwar. "Semi-supervised knowledge transfer for deep learning from private training data." International Conference on Learning Representations. 2017.
>
> [3] Tramer, F., & Boneh, D. Differentially private learning needs better features (or much more data). International Conference on Learning Representations. 2021.
>
> [4] Li, Xuechen, Florian Tramer, Percy Liang, and Tatsunori Hashimoto. "Large language models can be strong differentially private learners." International Conference on Learning Representations, Toulon, France. 2022.
>
> [5] Guo, Chuan, Alexandre Sablayrolles, and Maziar Sanjabi. "Analyzing Privacy Leakage in Machine Learning via Multiple Hypothesis Testing: A Lesson From Fano." *arXiv preprint arXiv:2210.13662*.
>
> [6] Balle, Borja and Cherubin, Giovanni and Hayes, Jamie. “Reconstructing Training Data with Informed Adversaries”. arXiv: 2201.04845.
>
> [7] Carlini, Nicholas and Liu, Chang and Erlingsson, Ulfar and Kos, Kernej and Song, Dawn. “The Secret Sharer: Evaluation and Testing Unintended Memorization in Neural Networks”.

---

### Public Comment · ~Zhiqi_Bu1 · 2022-11-15
**Deep concerns about the experiments**

Hi, I like the idea of group-wise clipping but I am concerned that the experiments are unfair and misleading. Please correct me if I miss anything. I put together a list of questions here, specifically for the per-layer clipping:

1. Per-layer clipping, even with adaptive thresholds, does not match the standard flat clipping. This renders your claim in the abstract "(Adaptive per-layer clipping) attaining similar or better task performance within *less wall time*" invalid. This is particularly obvious in Section 5.2 GLUE tasks. In Table 3, adaptive per-layer clipping uses **20 epochs** to get 92.40% accuracy but Li et al. (2022b) gets 92.09% using **3 epochs**. This comparison is confusing: you should either run adaptive per-layer clipping with 3 epochs to compare accuracy under the similar wall time, or run Li et al.'s method with 20 epochs for fair comparison.

2. Li et al. (2022b) results are **significantly under-reported in Table 3** for both Roberta-base and Roberta-large. I cite the original results from Table 1 (https://arxiv.org/pdf/2110.05679.pdf) here, for epsilon=3:
|                  | MNLI         | QQP    | QNLI   | SST2  |   |
|---------|----------|--------|--------|-------|---|
| full             | 82.47/82.10  | 85.41  | 84.62  | 86.12 |   |
| full + infilling | 82.45/82.99  | 85.56  | 87.42  | 91.86 |   |

The authors only cite the first row whereas the second row is SOTA. To clarify, the authors should highlight that infilling is not used in the main text and experiment adaptive per-layer clipping with infilling, instead of simply comparing to a weaker baseline.

3. The results in **Table 3 and Table 4 are contradicting each other**: for SST2 and Roberta-base, the adaptive per-layer with 20 epochs gets 92.03/92.40% in Table 3 but 91.57/91.96 in Table 4.

4. The methods compared in Table 3 are not of the same category.  In Table 3, Yu et al. (2021b), Li et al. (2022b), and adaptive per-layer train 100% model parameters, whereas Yu et al. (2022) is the only method that trains a subset of parameters using DP LoRA. I am not sure the comparison is meaning full. What about running per-layer clipping for DP LoRA? This should at least be clarified so readers can understand the difference.

5. For per-device clipping, the authors are using DP LoRA to train only a small fraction of 175B GPT3. Can the authors state how many trainable parameters are optimized? My estimate is that only millions of parameter are trained so the contribution may be less significant than it seems.

---

> ### Author Response · Authors · 2022-11-18
> **Answers to questions; concerns are most likely due to misunderstandings**
>
> Hi Zhiqi, thanks for bringing up these questions. While we agree that there were points of confusion in our previous draft (which we have clarified in the latest version), we genuinely don’t believe our experiments are “unfair and misleading,” as you suggested. We think that there likely are misunderstandings.
> Below we first clarify the major confusion points and thereafter answer each of your questions in detail. More details can be found in the updated draft (in particular, see Appendix G, Appendix H, the updated Section 5.2, and the updated Section 4).
>
> **Confusion point 1: The goal of Table 3 on GLUE tasks.**
>
> This is a major point of confusion and relates to your questions 1, 2, 3, and 4.
>
> First, note that these GLUE tasks have been used in several works in the DP deep learning literature, and thus there are numerous algorithms that report results on them. **Before touching on the issue of wall time, we wanted to understand the privacy-utility tradeoff of DP-SGD with adaptive per-layer clipping and compare it with other algorithms in the literature in this regard.** Since if adaptive per-layer clipping were not competitive at all on this front, it’d be questionable as to whether one should further study or adopt it.
>
> In other words, the goal of this experiment is to understand whether adaptive per-layer clipping (with reasonable hyperparameters) experiences comparable privacy-utility tradeoffs as other algorithms in the literature. Judging from the results in Table 3, we believe we have accomplished the target research goal.
>
> **Confusion point 2: text-infilling**
>
> Note that Li et al. 2022b demonstrated that the performance of private fine-tuning for text classification with various algorithms improves with a text-infilling formulation. **The text-infilling technique, however, reformats the dataset and reformulates the optimization problem so that learning is generally easier. The technique is orthogonal to the algorithmic aspects under study in this paper.** To be clear, none of the GLUE experiments performed in this paper adopts the infilling technique, as our goal was to strictly compare different algorithms on an equal footing (without being confounded by the format of the dataset or optimization problem). More generally, comparing method A without infilling against method B with infilling would unfairly disfavor method A.
>
> With the general issues settled above, we now address the specific questions.
>
> **Q1**
>
> > “Per-layer clipping, even with adaptive thresholds, does not match the standard flat clipping.”
>
> This is not true according to our experiments. The experiment in Table 4 is a carefully controlled study that compares adaptive per-layer clipping against flat clipping **when both methods fine-tune the same set of parameters and are run until a given epoch constraint.** Across the three different training epochs setups (E=10, 20, 30), we see that adaptive per-layer clipping has performance comparable to flat clipping with different privacy budgets (epsilon=3, 8). In addition, we included in Appendix G additional experiments on text generation that directly measure wall time improvement.
>
> > “This is particularly obvious in Section 5.2 GLUE tasks. In Table 3, adaptive per-layer clipping uses 20 epochs to get 92.40% accuracy but Li et al. (2022b) gets 92.09% using 3 epochs.”
>
> As mentioned above in confusion point 1, the goal of the experiment for Table 3 does not touch on wall time yet. That’s the goal of experiments in Table 4.

---

> > ### Author Response · Authors · 2022-11-18
> > **further comments**
> >
> > **Q2**
> >
> > > “Li et al. (2022b) results are significantly under-reported in Table 3 for both Roberta-base and Roberta-large.”
> >
> > We did not under-report their results. As stated in confusion point 2, none of the text classification workflows in our work adopted the text infilling method since we wanted to compare different algorithms on an equal footing without being confounded by the format of the datasets. **Thus, we reported the results in their paper obtained without infilling.** These numbers are generally a few percentage points below those obtained with infilling. We have clarified this in the main text.
> >
> > > “The authors only cite the first row whereas the second row is SOTA. To clarify, the authors should highlight that infilling is not used in the main text and experiment adaptive per-layer clipping with infilling, instead of simply comparing to a weaker baseline.”
> >
> > We stress that the overall research goal of Section 5.2 is not about obtaining the best performance possible without carefully considering what conditions give rise to those performances. Frankly, there are a handful of techniques that may lead to that (e.g., leveraging additional public data, better dataset formats, etc.), but this is beside the point.
> >
> > Our goal is instead to understand the privacy-utility tradeoff and computational efficiency of adaptive per-layer clipping as a primitive in DP learning algorithms. In other words, this work focuses on understanding the algorithmic aspect (not the data format aspect).
> >
> > **Q3**
> >
> > > “The results in Table 3 and Table 4 are contradicting each other: for SST2 and Roberta-base, the adaptive per-layer with 20 epochs gets 92.03/92.40% in Table 3 but 91.57/91.96 in Table 4.”
> >
> > Table 3 and Table 4 do not contradict each other. The results in Table 3 are based on fine-tuning all parameters. Results in Table 4 is based on fine-tuning only the attention and classification head layers. These details were already documented in Appendix A.2 in the original submission.
> >
> > In addition, we repeated the experiment in Table 4, but this time fine-tuning all parameters with both clipping approaches. Results in Appendix H (Table 12) show that adaptive per-layer clipping attains comparable or better performance than flat clipping when the number of training epochs is constrained to one of 3, 10, 20, and 30.
> >
> > **Q4**
> >
> > >  “The methods compared in Table 3 are not of the same category. In Table 3, Yu et al. (2021b), Li et al. (2022b), and adaptive per-layer train 100% model parameters, whereas Yu et al. (2022) is the only method that trains a subset of parameters using DP LoRA. I am not sure the comparison is meaning full.”
> >
> > As elaborated in confusion point 1, the goal of the experiment in Table 3 is to understand the privacy-utility tradeoff of DP-SGD with adaptive per-layer clipping and show that it is not systematically worse than approaches in the literature.
> >
> > **Q5**
> >
> > > “For per-device clipping, the authors are using DP LoRA to train only a small fraction of 175B GPT3. Can the authors state how many trainable parameters are optimized?”
> >
> > The number of parameters being optimized is ~151 million. This is because we set the LoRA rank to 32 (future work may study how things change as a function of the LoRA rank). Calculation as follows:
> >
> > 12288 (d_model) x 32 (LoRA rank) x 2 (LoRA layers/linear layer) x 96 (Transformer blocks) x 2 (linear layers/Transformer block) approx. = 151 million.
> >
> > We believe that the results we obtained on summarization with GPT-3 are quite significant. We note that the per-device clipping method was only a means to achieve this end goal—- it is not the end goal itself.
> >
> > Nevertheless, we stress that the parameter count doesn’t reveal the full story for fine-tuning with DP models that don’t fit on single devices. In particular, synchronizing all devices after computations for a micro-batch finish (to clip gradient) is costly not because the all-gather operation is costly in itself but rather because it disrupts the schedule of pipeline parallelism (we encourage you to read our draft on the brief intro to pipeline parallelism). We stated the technical reasons in the original draft. During the rebuttal period, we revised our draft (Section 4) to elaborate further. Please take a read of our draft if you are interested in the specifics.
> >
> > Again, as stated in the draft, this is “the first attempt at experimenting with DP fine-tuning on huge models.” We’re optimistic DP fine-tuning will become easier with such models with improvements in distributed training. But at the moment, per-device clipping---due to its simplicity, efficiency, and composability with pipeline schedules---is our go-to approach for dealing with such models.

---

### Author Response · Authors · 2022-11-19
**thank you for the reviews and feedback; summary of our updates**

We thank the reviewers as well as non-reviewers for their reviews and valuable feedback on our draft.

We included detailed responses to each post below. Here, we summarize the updates in the latest draft.

- We included additional visualizations demonstrating the wall time improvement of adaptive per-layer clipping in the new Appendix G.
- We elaborated further in Section 4 on the subtleties of applying DP-SGD to fine-tune models like GPT-3 (which requires multiple devices to host due to scale) based on pipeline parallelism. The key challenge we'd like to emphasize is that clipping the entire gradient requires communication that disrupts the pipeline schedule that coordinates devices and micro-batches (within a minibatch); per-device clipping helps bypass this complication by avoiding this communication.
- There were additional concerns raised by a non-reviewer about the experiments in Section 5.2. We have edited Section 5.2 to clarify issues of confusion further (see comments addressed to this non-reviewer for details). In addition, we included in the new Appendix H additional experiments to validate our claim that adaptive per-layer clipping attains utility competitive with flat clipping under fixed training epoch and privacy budget constraints.
- We included clarifications regarding the selection of the privacy budget in Section 5.

Finally, we note that we will release all source code for reproducing the GLUE experiments, the GPT-2 fine-tuning experiments, and the CIFAR-10 experiments after the review period. While we are unable to release code for the GPT-3 experiments due to policy issues, we will make our best effort to provide guidelines for a re-implementation and document potential pitfalls to ease researchers who would like to reproduce our results with a publicly available model.

---

### Decision · Program_Chairs · 2023-01-20

**Decision:**

Accept: poster

**Justification For Why Not Higher Score:**

The paper basically applied existing DP techniques. The novelty is fairly limited but the experiments are interesting and convincing.

**Justification For Why Not Lower Score:**

The experiments are comprehensive, interesting, and convincing. The paper is well-written and will be a good addition to the literature of applying DP for large-scale deep learning.

**Metareview: Summary, Strengths And Weaknesses:**

This paper applied existing differential privacy (DP) techniques for fine-tune large-scale language models GPT2 and GPT3. They utilized per-layer gradient clipping with adaptively tuned clipping bounds  which are shown to be  efficient and achieve similar utility as flat clipping. Reviewers agree that the paper is largely well-written and the experiments are comprehensive and convincing. The paper attracted additional comments from a non-reviewer reader and the authors' response appear satisfactory.

**Note From Pc:**

if the above contains the word "oral" or "spotlight" please see: "oral" presentation means -> notable-top-5% and "spotlight" means -> notable-top-25%. As stated in our emails, we are disassociating presentation type from AC recommendations